# Effects of different reference periods on drought index (SPEI) estimations from 1901-2014

Myoung-Jin Um[1], Yeonjoo Kim[1,*], Daeryong Park[2], Jeongbin Kim[1]

[1]Department of Civil and Environmental Engineering, Yonsei University, Seoul, 03722, Republic of Korea
[2]Department of Civil, Environmental and Plant Engineering, Konkuk University, Seoul 05029, Republic of Korea

*Correspondence to*: Yeonjoo Kim (yeonjoo.kim@yonsei.ac.kr)

**Abstract**. This study aims to understand how different reference periods (i.e., calibration periods) of climate data used to estimate drought indices influence regional drought assessments. Specifically, we investigate the influences of different reference periods on historical drought characteristics, such as the trend, frequency, intensity and spatial extent, using the standardized precipitation evapotranspiration index (SPEI) with a 12-month lag (SPEI-12), which was estimated from the datasets of the Climate Research Unit (CRU) and the University of Delaware (UDEL). For the 1901–1957 (P1) and 1958–2014 (P2) estimation periods, three different types of reference periods are used to compute the SPEI: P1 and P2 together, P1 and P2 separately and P1 only. Focusing on East Asia, Europe, the United States and West Africa, we find that the influence of the reference period is significant in East Asia and West Africa, with dominant drying trends from P1 to P2. The reference period influenced the assessment of drought characteristics, particularly the severity and spatial extent, whereas the influence on the frequency was relatively small. Finally, self-calibration, which is the most common practice for indices such as the SPEI, tends to underestimate the drought severity and spatial extent relative to the other approaches used in this study. Although the conclusions drawn in this study are limited by the use of two global datasets, they highlight the need for clarification of the reference period in drought assessments to better understand regional drought characteristics and the associated temporal changes, particularly under climate change scenarios.

## 1 Introduction

Drought is a complex, slow onset and natural phenomenon that affects more people than any other hazard and seriously influences water resources, agriculture, society and ecosystems (Hagman, 1984; Wilhite, 2002; Ionita et al., 2015). Because drought impacts are largely nonstructural and spread over relatively large regions, the onset and end of a drought, as well as its severity, are often difficult to determine (Wilhite, 2002). Furthermore, based on recent changes in the 21st century and projected climate warming, such drought phenomena will likely worsen (Sheffield and Wood, 2008; Dai, 2011a). Sheffield et al. (2012) stated that severe and prolonged drought events have been observed since the 1970s, and these changes are related to higher temperatures and lower precipitation.

Drought can be defined and explained using absolute or relative terminology, which allows terms and measures to be compared (Dai, 2011b; Trenberth et al., 2014). Absolute terms include the amount of precipitation, the

amount of soil moisture and other metrics. The relative measures include the Palmer drought severity index (PDSI), the standardized precipitation index (SPI), the standardized precipitation and evapotranspiration index (SPEI) and others. Relative drought indices, however, are limited in their utility because they are based on standardized or normalized shortages relative to average conditions at a given station or in a specific period (Vicente-Serrano and Beguería-Portugués, 2003; Vicente-Serrano et al., 2010). Nevertheless, various drought indices have been widely used in many drought studies.

Dracup et al. (1980) suggested three components of drought: duration, magnitude (average water deficiency) and severity (cumulative water deficiency). Such concepts have been applied to various drought indices to analyze historical characteristics. Wang et al. (2011) defined the intensity-duration-frequency of droughts with the SPI, standardized runoff index (SRI), standardized soil water index (SSWI) derived from observations and future regional climate change projections in central Illinois. To evaluate how well global climate models simulated observed drying or wetting trends, Nasrollahi et al. (2015) applied the Mann-Kendall trend test to SPIs derived from global observational climate data, in this case, the dataset from the Climate Research Unit (CRU), and 41 predictions of global climate models (GCMs) from the Coupled Model Intercomparison Project Phase 5 (CMIP5). Similarly, Tan et al. (2015) utilized climate data from 22 meteorological stations in Ningxia, a well-known food production area in Northwest China, and performed Mann-Kendall trend tests with the SPI and SPEI. The degrees of increasing drought frequency and intensity varied with the stations in the study region. Furthermore, Touma et al. (2015) used data from 15 GCMs in CMIP5 and assessed the likelihood of changes in the spatial extent, duration and number of occurrences of four drought indices, including the SPI, SPEI, and others. Zhao and Dai (2015; 2016) assessed the self-calibrated PDSI (sc-PDSI) with multiple CMIP3 and CMIP5 model projections at the globe scale and showed that the drought frequency and area increased with increasing sc-PDSI, even under low to moderate emission scenarios.

Estimating a drought index requires a calibration step. Specifically, historical data such as precipitation data should be fitted to a specific probability distribution function (PDF) and used to estimate drought indices. Some previous studies have addressed the issue of the data period in the calibration step (e.g., Karl et al., 1996; Dubrovsky et al., 2009; Dai and Zhao, 2016). While it is common to use self-calibrated indices (i.e., using the same dataset for calibration and index estimation), some studies have proposed calibration using reference climate data to allow for an intercomparison of the index among stations or different periods (Dubrovsky et al., 2009). Such reference periods (i.e., calibration periods) of climate data are particularly important in climate change studies. It was previously noted for the sc-PDSI that trends toward more extreme conditions are amplified when the calibration period does not include recent data, including the recent effects of climate change (van der Schrier et al., 2013; Trenberth et al., 2014). Dai and Zhao (2016) examined uncertainties in the sc-PDSI due to different choices of forcing data and the calibration period. They recommend using the Global Precipitation Climatology Center (GPCC) or the Global Precipitation Climatology (GPCP) datasets over other existing land precipitation products, such as CRU, and not including years after 1980 in the calibration period due to the influence of anthropogenic climate change. Still, few studies have clarified their approaches to calibration.

Therefore, in this study, we aim to understand how a different reference period (i.e., calibration period) of climate data influences regional drought assessment. Specifically, we investigate the influences of different reference periods on historical drought characteristics, such as the trend, frequency, intensity and spatial extent,

with the SPEI estimated using two historical global climate datasets from the CRU and the University of Delaware (UDEL). Our study shows that the reference period influences the assessment of drought characteristics, particularly the severity and spatial extent, while its influence on the frequency is relatively small. These influences are especially significant in regions with dominant drying trends, such as East Asia and West Africa. These findings suggest that the reference period should be clarified in drought assessments for a better understanding of regional drought characteristics and their temporal changes.

## 2 Data and method

### 2.1 Study area and climate data

We investigate the drought characteristics in the Northern Hemisphere with a focus on four different regions: East Asia, Europe, the United States and West Africa (Fig. 1). We performed analyses based on the spatially distributed patterns in those regions, as well as the average trends, but without distinguishing subregions based on climate characteristics. Two widely used global observational datasets from the CRU and UDEL are utilized in this study. Specifically, monthly precipitation and temperature data from 1901 to 2014 with a spatial resolution of 0.5° are used.

This study uses the latest CRU dataset (CRU TS3.23), as described in Harris et al. (2014). The principal sources of the CRU data are the World Meteorological Organization (WMO) in collaboration with the US National Oceanographic and Atmospheric Administration (NOAA). Covering all land areas between 60°S and 80°N at a spatial resolution of 0.5°, the dataset includes global monthly climate data for ten variables: precipitation, mean temperature, diurnal temperature range, minimum and maximum temperature, vapor pressure, cloud cover, rain days, frost days and potential evapotranspiration. The dataset is derived from archives of climate station records with extensive manual and semi-automated quality control measures.

The UDEL dataset (V 4.01, Willmott and Matsuura, 2001) is also used in this study. The dataset includes gridded monthly precipitation and temperature data at a spatial resolution of 0.5° and a global scale. The dataset was compiled from sources including the Global Historical Climatology Network (GHCN) and the Global Surface Summary of the Day (GSOD). To interpolate the station values to the grid, climatologically aided interpolation (CAI) and traditional interpolation were used for precipitation, and digital elevation model (DEM)-assisted interpolation, traditional interpolation and CAI were used for temperature. In this work, traditional interpolation is based on a spherical version of Shepard's algorithm, which employs an enhanced distance-weighting method (Shepard, 1968; Willmott et al., 1985).

As briefly noted in Section 1, Dai and Zhao (2016) suggested that the GPCC or GPCP dataset should be used instead of the CRU datasets in drought assessment with the sc-PDSI. They noted the limitations of the CRU dataset (specifically CRU TS3.10.01) due to its poor data coverage since the 1990s. In this study, the CRU TS3.23 and the UDEL datasets are used because these datasets provide both precipitation and temperature data, whereas the GPCC and GPCP datasets include only precipitation data.

## 2.2 Meteorological drought index

Various drought indices have been used to understand different types of droughts, including meteorological drought, agricultural drought and hydrological drought (Heim, 2002). For meteorological droughts, the indices include the PDSI (Palmer, 1965), the SPI (McKee et al., 1993) and the SPEI (Vicente-Serrano et al., 2010). As different studies have used different meteorological drought indices (Seneviratne, 2012; Sheffield et al., 2012; Trenberth et al., 2014; Nasrollahi et al., 2015; Touma et al., 2015), this study focuses on the SPEI. Devised by Vicente-Serrano et al. (2010), the SPEI has the advantage of considering the effects of temperature variability on drought relative to the SPI (Naumann et al., 2014). The SPEI uses the amount of precipitation minus PET and fits the data to the log-logistic PDF. Here, we summarize the steps in estimating the SPEI based on monthly precipitation and temperature data. The detailed procedure for estimating the SPEI was presented by Vicente-Serrano et al. (2010).

Step 1: Estimate the water surplus or deficit in month j ($D_j$) using the difference between precipitation ($P_j$) and potential evapotranspiration ($PET_j$).

$$D_j = P_j - PET_j \qquad (1)$$

Here, the potential evapotranspiration is estimated based on the Thornthwaite method (1948), which requires the monthly temperature, latitude, day and month.

Step 2: Estimate the cumulative difference ($X_{i,j}^k$) over timescale $k$ in a given month $j$ and year $i$. For example, the cumulative difference for a month in a particular year based on a 12-month timescale can be calculated as follows.

$$X_{i,j}^k = \sum_{l=13-k+j}^{12} D_{i-1,l} + \sum_{l=1}^{j} D_{i,j}, \qquad if\ j < k \qquad (2)$$

$$X_{i,j}^k = \sum_{l=j-k+1}^{j} D_{i,l}, \qquad if\ j \geq k \qquad (3)$$

Step 3: Fit the cumulative difference to a log-logistic distribution as follows:

$$F(X) = \left[1 + \left(\frac{\alpha}{x-\gamma}\right)^{\beta}\right]^{-1} \qquad (4)$$

where $F(X)$ is the cumulative probability function of a three-parameter log-logistic distribution and $\alpha, \beta$ and $\gamma$ represent the scale, shape and origin parameters, respectively. For model fitting, the L-moment procedure (Hosking, 1990) is employed, as it is one of the most robust and easy-to-use approaches.

Step 4: Estimate the SPEI based on the estimated $F(X)$. The SPEI can be derived from the standardized values of $F(X)$ and the classical approximation of Abramowitz and Stegun (1965) following Vicente-Serrano et al. (2010). The estimated drought index is classified as shown in Table 1 for moderate, extreme and very extreme cases. In this study, we focused on the SEPI with a 12-month lag (SPEI-12). SPEI can be estimated for different lag times, such as 1, 3, 6, 9, 12 and 24 months.

## 2.3 Temporal trends and statistical characteristics

This study investigates various measures of historical droughts, including the trend, frequency, severity and spatial extent (Lloyd-Hughes and Saunders, 2002; Wang et al., 2011; Hoerling et al., 2012; Seneviratne, 2012; Trenberth et al., 2014; Touma et al., 2015).

The temporal trend is investigated with a nonparametric and monotonic trend test based on the S-statistic of the Mann-Kendall trend test (Mann, 1945; and Kendall, 1976). In this test, an increasing (positive) trend or decreasing (negative) trend is tested for at a significance level of 5%. Different measures have been defined and used in past studies to assess the frequency, severity and spatial extent of drought (e.g., Wang et al., 2011; Touma et al., 2015) because it is not straightforward to define these quantities in practice. For example, Touma et al. (2015) defined the duration, occurrence and spatial extent of drought to investigate the drought changes with 15 CMIP5 models throughout the world in the 21st century. The duration of drought was defined as the consecutive period below a certain drought threshold. The occurrence of drought was defined as the total number of droughts in the period of interest. Additionally, the spatial extent of drought was defined as the percentage of grid points below the given drought level, in which the corresponding drought index was less than the given drought category in each month relative to the total number of terrestrial grid points in the domain.

In this study, we defined three measures of drought based on the SPEI-12: (1) drought frequency was calculated as the ratio of the total number of drought events (i.e., SPEI-12 $\leq$ -1) to the total number of terrestrial grid points; here, we counted the number of drought events without considering whether a given drought event (i.e., SPEI-12 $\leq$ -1) was identified consecutively. (2) Severity was defined as the lowest estimate of the regional monthly average SPEI-12 using moving windows with periods of 1 to 12 months; here, regional averages were estimated in the four study regions depicted in Fig. 1. (3) The spatial extent was calculated as the number of grid points with an annual SPEI-12 $\leq$ -1.0 relative to the total number of terrestrial grid points.

**2.4 Design of data analysis**

To understand the influence of the reference period (i.e., calibration period) on the drought index, three different types of reference periods are used to estimate the SPEI-12 with the CRU and UDEL data. To separately analyze the drought characteristics in the periods of 1901–1957 (P1) and 1958–2014 (P2), different reference periods are used (Table 2). Here, we assume that the mean climates of P1 and P2 are different to some extent because of global climate and environmental changes, which will be discussed further in Section 3. For the first type of reference period (Ref1), we calibrated the distribution of a specific PDF (Step 3 in Section 2.2) using data from 1901 to 2014, which is used to estimate the SPE12 for the P1 and P2 estimation periods. For the second type of reference period (Ref2), calibrations are performed separately for P1 and P2; thus, so-called self-calibrated indices are derived. For the third type (Ref3), we calibrated the distribution using the data from P1 (i.e., 1910–1957) and then use this distribution for both estimation periods.

**3 Results and discussion**

**3.1 Spatial and temporal patterns of climate variables**

In this section, we examine the spatial and temporal variations in precipitation, air temperature and PET (Figs. 2, 3 and 4 and Table 3), which are used to estimate D (= P-PET) (in Eq. 1) and the SPEI values. We particularly

focused on the differences in meteorological conditions between P1 and P2 to enhance our understanding of similar or different drought index values according to the different reference periods in the following sections.

To investigate the temporal changes in precipitation, air temperature and PET, we compared the means and standard deviations between the two periods (i.e., P1 and P2) (Figs. 2 and 3 and Table 3). Most cases showed largely consistent results between CRU and UDEL; therefore, we did not focus extensively on the differences between the two datasets. In general, the temporal pattern of precipitation varied among regions, and air temperature increases were observed in all regions. On average (Fig. 3 and Table 3), annual precipitation decreased in P2 relative to P1, as in East Asia and West Africa, whereas decreases in precipitation were only evident in limited areas within the regions (Fig. 2); for example, the west Sahel within West Africa. In contrast, annual precipitation increased in Europe and the United States. Increases in air temperature were clearly shown in all regions; consequently, increases in PET, which is controlled mainly by air temperature, were generally evident. Decreases in D were observed only in East Asia and West Africa (Fig. 4c). In these regions, an annual water deficit (i.e., negative D) was evident, whereas in other regions, i.e., Europe and the United States, an annual water surplus (i.e., positive D) was observed.

The Mann-Kendall trend tests were also performed for annual precipitation, annual average temperature and annual PET, as shown in Fig. 4. The data reflect whether these variables showed statistically increasing, statistically decreasing or no trends. For annual precipitation in EA, the areal extent with an increasing trend was almost twice that with a decreasing trend based on CRU, but the areal extent with a decreasing trend based on UDEL was broader than that with an increasing area. In Europe and the United States, the areal extent of an increasing trend was clearly larger than that of a decreasing area based on both CRU and UDEL. However, in WA, the areal extent of a decreasing trend was larger than that of an increasing trend based on both CRU and UDEL. These patterns were generally more severe for CRU than for UDEL. For annual average air temperature and PET, CRU produced increasing trends in most regions. Similar patterns were observed for UDEL, but the areal extent of the decreasing trend was slightly larger than that of CRU.

**3.2 Temporal patterns of the drought index**

The drought index (i.e., SPEI-12) was estimated by fitting the three-parameter log-logistic model based on three different reference periods (Table 2), as described in Section 2.4. As shown in the L-moment ratio diagram with the CRU data and Ref1 as an example (Fig. 5), the model is well fit by the L-moment approach, following Vicente-Serrano et al. (2010). Fig. 6 shows the temporal variations in SPEI-12 based on the reference periods (Ref1, Ref2 and Ref3) and datasets (CRU and UDEL) used in the two periods. In the United States and Europe, the SPEI-12 averages are very similar in the two periods, with values of 0.005 (P1) and 0.118 (P2) in the United States and -0.011 (P1) and -0.001 (P2) in Europe. In East Asia, the SPEI-12 averages in the three different reference periods slightly decrease from P1 to P2, whereas the deviations in SPEI-12 increase markedly. In West Africa, the averages and deviations in SPEI-12 significantly decrease and increase, respectively, from P1 to P2. Furthermore, the variances in SPEI (box lengths in Fig. 5) are relatively small in P1 compared with those in P2 in East Asia and West Africa, whereas no noticeable differences in the variances are observed in Europe and the United States. This result may be attributed to the lack of ground-based observations before 1950 (i.e., most of

P1). As suggested in previous studies (i.e., Becker et al., 2013; Vittal et al., 2013; Nasrollahi et al., 2015), the limited availability of data in the early 20$^{th}$ century can result in underestimates of the spatial variabilities of climate variables in global datasets; in the present study, such limited data availability might have contributed to the reduced SPEI variance in P1 in East Asia and West Africa. Based on regional averages, the role of the reference period is not clear; thus, we investigate the spatial patterns of SPEI-12 hereafter.

Based on the Mann-Kendall trend test of annual SPEI-12 from 1901 to 2014, we identified the areas with increasing (i.e., wetting), decreasing (i.e., drying) and no trends in each region (Fig. 7). First, the spatial distribution of SPEI-12 trends is identical between Ref1 and Ref3, and that in Ref2 is different. Ref1 and Ref2 use different calibration datasets but are similar in using one dataset for the two estimation periods; however, Ref2 uses different calibration datasets for different estimation periods (Table 4). Therefore, the SPEI-12 of Ref2 exhibits relatively small areas of wetting and drying trends in the first and second periods relative to those of Ref1 and Ref3.

Regarding the temporal characteristics in different regions, our findings for Ref1 and Ref3 are as follows. In WA, drying trends are clearly dominant. In EU, drying trends are scattered over the domain. In the US, wetting trends are scattered in the eastern region, and drying trends can be observed in the southwestern region. In East Asia, the drying trends are clearly in the western region.

Based on the grid-level trend analyses of precipitation, air temperature, PET and SPEI-12, we categorized each grid cell based on increasing, decreasing or neutral trends for each variable (Fig. 8). For SPEI-12, increasing and decreasing trends represent wetting and drying trends. We present the ratio of each case relative to the total number of cases (i.e., total number of terrestrial grid cells in all four regions). First, the SPEI-12 trends are the same between Ref1 and Ref3, as the estimation periods share one reference period in both Ref1 and Ref3, while each estimation period uses its own reference period in Ref2. Thus, the values of SPEI-12 are different in both cases, but the trends (i.e., relative values) are the same. Second, precipitation and air temperature exhibit neutral (or no) trends (i.e., the center panel among the 3 x 3 panels in Fig. 8, indicating a presumably stationary climate), and the grid percentages of different trends in SPEI-12 vary between Ref1/Ref3 and Ref2. However, the ratio is relatively small, as most grid cells display increasing temperature and PET trends. Finally, in the case of neutral precipitation and increasing air temperature (or PET) trends (i.e., the top middle panel among the 3 x 3 panels in Fig. 8), the numbers of cells with neutral and drying SPEI-12 trends are notably different between Ref1/Ref3 and Ref2. We observed increasing temperature and thus increasing PET trends in most regions (refer to Fig. 4). This discrepancy between the reference periods might play a critical role in assessing the drought status.

**3.3 Frequency, severity and spatial extent of drought**

In this section, we examine how the reference periods play a role in assessing the frequency, severity and spatial extent of drought using SPEI-12. The definitions of frequency, severity and spatial extent of drought used in this study are clarified in Section 2.3, and they may differ in different studies.

As explained above, a drought event occurs when the monthly SPEI-12 is estimated to be at or below -1.0 based on the drought duration-frequency relationship. For each drought event in a grid cell, the duration is how long the SPEI-12 stays at or below -1. The frequency is the ratio between the total number of drought events and the

number of terrestrial grid points in each region (Fig. 9). We found that the drought events with long durations (prolonged right tails in the plot) occur more frequently in P2 than in P1 in all regions. However, we did not find any particular differences between the three different reference periods except in WA. The drought frequencies differ among the three reference periods. The frequencies of Ref2 and Ref3 are higher than those of Ref1 in P1, and slight differences in the frequency among the three reference periods are observed throughout the 12-month duration of P2.

We examine how the severity of drought varies with the moving window size for the average monthly SPEI-12. Fig. 10 shows the most severe SPEI-12 estimates, which are defined as the lowest values among the regional monthly averages of SPEI-12 in the moving windows from 1 month to 12 months. In Europe and the United States, we found no large differences between the SPEI-12s of Ref1, Ref2 and Ref3 in the same period. In these regions, the most severe SPEI-12s in P1 are higher than those in P2. Such findings are seemingly inconsistent with the recently observed severe drought events in the United States and Europe, but they are reasonable because we examined the regionally averaged indices and not the local extremes of SPEIs. Additionally, the results are consistent with Fig. 3c. In the United States (the third row of Fig. 3c), the increase in precipitation is higher than that in PET, which increases D (Eq.1). In Europe (the second row of Fig. 3c), the increase in PET is higher than that in precipitation; thus, D decreases on average. However, at the lower extreme of D in this case (i.e., the lower extent of the vertical line in the box plot of D in Fig. 3c), a slight increase is apparent, indicating that the most severe drought events are less severe in P2 than in P1. By examining the spatial maps of the most severe cases (not shown), we found that severe drought events in P1 were more widespread than in P2. Such widespread drought might be due to the sparse network of meteorological stations during the early 20th century, a possibility that requires further study.

In East Asia and West Africa, different patterns can be observed for the most severe SPEI-12 values. The annual precipitation and air temperature (and thus PET) exhibit regionally scattered decreases and widespread increases, respectively (Fig. 4). Consequently, the droughts from 1958–2014 are more severe than those in P1. Furthermore, the severity varies significantly with the calibration period in East Asia and West Africa, and the changes in precipitation and air temperature between the two periods are considerable.

The spatial extents of droughts for annual SPEI-12 ≤ -1.0 are examined by sorting the results in ascending order (Fig. 11). We count the numbers of grid points with SPEI-12 values less than -1.0 in each period (i.e., P1 and P2) and divide them by the number of terrestrial grid cells in the region to derive the spatial extent, i.e., the grid-based percentage of droughts. Then, the annual time series of the spatial extent are sorted in ascending order. No specific patterns are evident in Europe and the United States. In East Asia and West Africa, the spatial extents are generally broader in P2 than in P1. Notably, the spatial extents from 1958–2014 clearly diverge based on the different calibration periods, reflecting the importance of the calibration method (i.e., the reference period used to assess the droughts in a region).

To understand how the drought characteristics change if the reference period is dry or wet, we compared the spatial extent of drought (%) for dry and wet cases in four regions. We defined dry and wet cases based on the water surplus or deficit D (Eq. 1). Then, we compared D values between the reference period and estimation period. A value of D in the estimation period less than that in the reference period represents a dry case, i.e., the estimation period is drier than the reference period. We performed such analyses only in Ref1 for the estimation

periods of 1901-1957 (P1) and 1958-2014 (P2), as well as a reference/calibration period from 1901-2014 (P1+P2). For dry and wet cases, we quantified the spatial extent (%) according to the three different drought levels (D1, D2 and D3, which denote the cases of SPEI<-1.0, SPEI<-2.0 and SPEI<-3.0, respectively) in the four regions.

As presented in Table 5, the average D in P1 and P2 (estimation period) is smaller than that in P1+P2 (reference period), and these are considered dry cases. For example, in EA, the D values in P2 and P1+P2 are -4.89 mm/month and -5.07 mm/month, respectively; thus, these are dry cases. Then, for each case, the spatial extent of drought, i.e., the number of drought grid cells relative to the total number of terrestrial grid cells, is analyzed, as shown in Fig. 12. The spatial extent of drought tends to be larger in dry cases than in wet cases in most regions,
particularly in West Africa. However, we also noted that there are a few exceptions to this trend that may be attributed to the fact that we used regionally averaged values of D. Thus, we cannot consider the grid-level variability in D values.

### 3.4 Case studies using historical drought events

SPEI-12s with different reference periods are evaluated for historical drought events selected in each region to investigate how different reference periods influence the drought assessments of historical events. One drought event is chosen in each region as follows: 1) in East Asia, droughts that occurred in northern China in 2001 are chosen, and these events caused economic losses of USD 1.52 billion (Zhang and Zhou, 2015); 2) in EU, we chose a 2003 drought that was caused by the European heat wave and spread over the majority of Europe (Stagge
et al., 2013; Spinoni et al., 2015); 3) in the United States, we chose 2012 as the period of study because a historically extensive drought occurred over half of the United States and caused economic losses of USD 31.2 billion (Smith and Katz, 2013; National Climate Data Center, 2015); and 4) in West Africa, the drought in 1984 was chosen because it was one of the most severe droughts that has occurred in Sahel countries (Gommes and Petrassi, 1994; Rojas et al., 2011; Masih et al., 2014).

By estimating SPEI-12 for a chosen year in each region, we can compare the magnitudes of SPEI values (Figs. 13, 14, 15 and 16). Here, the annual SPEI-12 values based on monthly climate data from January to December in each year are first calculated. Then, the SPEI-12 values of a chosen year are examined in detail. All SPEI-12 values in different reference periods reflect the drought status because we chose specific years with drought events. In general, all cases reveal that the SPEI-12 estimates in Ref2 are relatively high (i.e., wet), and those in
Ref3 are relatively low (i.e., dry) in East Asia and West Africa, where drying temporal trends are clear. In particular, several extreme values (i.e., out of the scale range in Figs. 13-16) of SPEI-12 in Ref3 cases highlight the importance of the reference period. If a reference period is based on a certain time (P1 in this study, i.e., Ref3), the drought events in the estimation period may be beyond the range in which the distribution is calibrated for the index. Essentially, for Ref3, it is assumed that not only the stationarity of the climate but also that the
entire probability distribution of droughts is sampled in this period.

Furthermore, the percentage of the spatial extent of drought, i.e., the number of drought grid points relative to the total number of grid points, is assessed for different drought thresholds (Table 6). In most cases, the spatial extents of drought with SPEI values less than a certain threshold, such as -1, -2 or -3 (i.e., D1, D2 and D3, as in

Table 1), are the greatest in Ref3 among the three cases with different reference periods. These results and the spatial extents are consistent with the SPEI-12 results estimated above. In addition, higher percentages of severe droughts events, which are defined based on low thresholds, such as SPEI-12 values less than -2 or -3, were observed in Ref3 than in Ref1 and Ref2 in all four regions of this study.

## 4 Conclusions

This study seeks to understand how a different reference period (i.e., calibration period) of climate data can influence drought index estimation and regional drought assessment. Specifically, we investigated the influences of different reference periods on historical drought characteristics, such as the trend, frequency, intensity and

spatial extent, using SPEI-12 and the CRU and UDEL datasets. For the 1901–1957 (P1) and 1958–2014 (P2) estimation periods, three different types of reference periods are used. In the first case, data from 1901 to 2014 (P1+P2) are used for both estimation periods. In the second case, data from P1 and P2 are used separately for the estimation periods of P1 and P2, respectively (self-calibrated). In the final case, data from P1 (1910–1957) are used for both estimation periods.

Focusing on the four selected regions of this study, i.e., East Asia, Europe, the United States and West Africa, we found that the influence of the reference period is significant in regions with dominant drying trends from P1 to P2, such as East Asia and West Africa. Additionally, the results suggest that it is necessary to quantify the trends of climate variables such as precipitation and air temperature as the first step in selecting a reference period. Our results also show that the reference period influences the assessment of drought characteristics, particularly the

severity and spatial extent, based on the two datasets; however, their influence on the frequency is relatively small. Finally, we found that the use of a distribution calibrated with recent observations (i.e., Ref1 and Ref2 with the calibration periods of 1901-2014 and 1958-2014, respectively) tends to underestimate the drought severity and spatial extent relative to another approach used in this study (i.e., Ref3 in Table 2 with the calibration period of 1901-1957).

These findings suggest that recent periods should potentially be excluded from the calibration to better understand recent drought events, particularly in regions such as East Asia and West Africa, where dominant drying trends are observed. This highlights the need for clarifying the reference period in drought assessments to better understand regional drought characteristics and their temporal changes. Such a clarification is particularly critical for assessing droughts under climate change scenarios and developing adaptation strategies for water

resource management in the context of climate change.

However, we note that the abovementioned results are drawn from only three sets of reference periods, two different datasets (i.e., CRU and UDEL) and four regional examples. Future work should evaluate different combinations of reference periods with increased sample sizes and different datasets. The combined datasets could also be used to focus on the effects of different precipitation or temperature products on the SPEI. For

example, the precipitation data from the CRU, UDEL and GPCC, which is suggested to be better than CRU data by Dai and Zhao (2016), and the temperature data from CRU could be utilized to focus on the effects of different precipitation products.

This study, which was based on historical data, may yield different results at the local scale, and similar studies based on historical data and climate change scenarios in different regions would undoubtedly strengthen our findings. In the present study, we focused on the temporal aspects of calibration data (i.e., the calibration period). As briefly mentioned in the Section 1, using data from a particular station or grid to obtain averaged data for calibration could permit a meaningful comparison of drought indices at different locations. In conjunction with temporal considerations, spatial issues should be addressed in future studies.

Furthermore, we noted that the Thornthwaite approach, in which air temperature is the main controlling factor of PET, is used to estimate the SPEI in this study; however, other approaches, such as the Penman method, could be used to consider changes in other meteorological variables, such as wind, atmospheric humidity and radiation. McVicar et al. (2012) suggested that temperature increases may have limited effects on drought through increased PET because other meteorological conditions that affect PET may compensate for the temperature increase.

## Acknowledgements

This study was supported by the Korea Meteorological Administration R&D Program under Grant KMIPA 2015-6180 and by the Basic Science Research Program through the National Research Foundation of Korea funded by the Ministry of Science, ICT & Future Planning (2015R1C1A2A01054800).

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

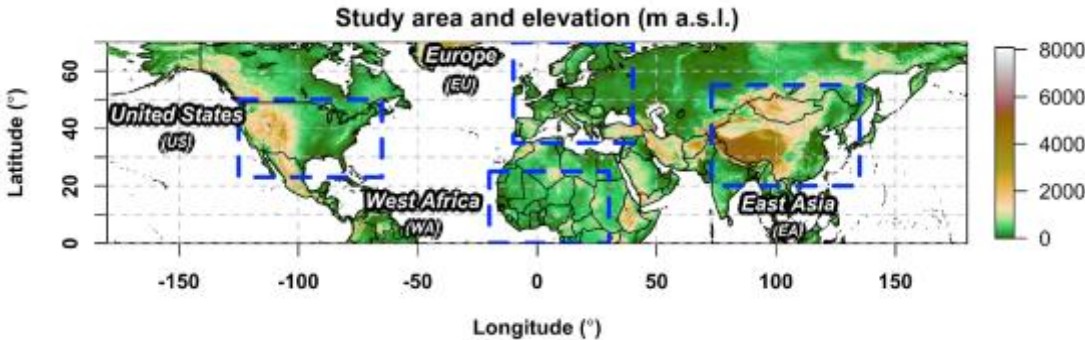

**Figure 1. Study area, including East Asia, Europe, the United States and West Africa, and elevation (m above sea level (a.s.l.)). The dashed blue boxes represent the boundaries of each study region.**

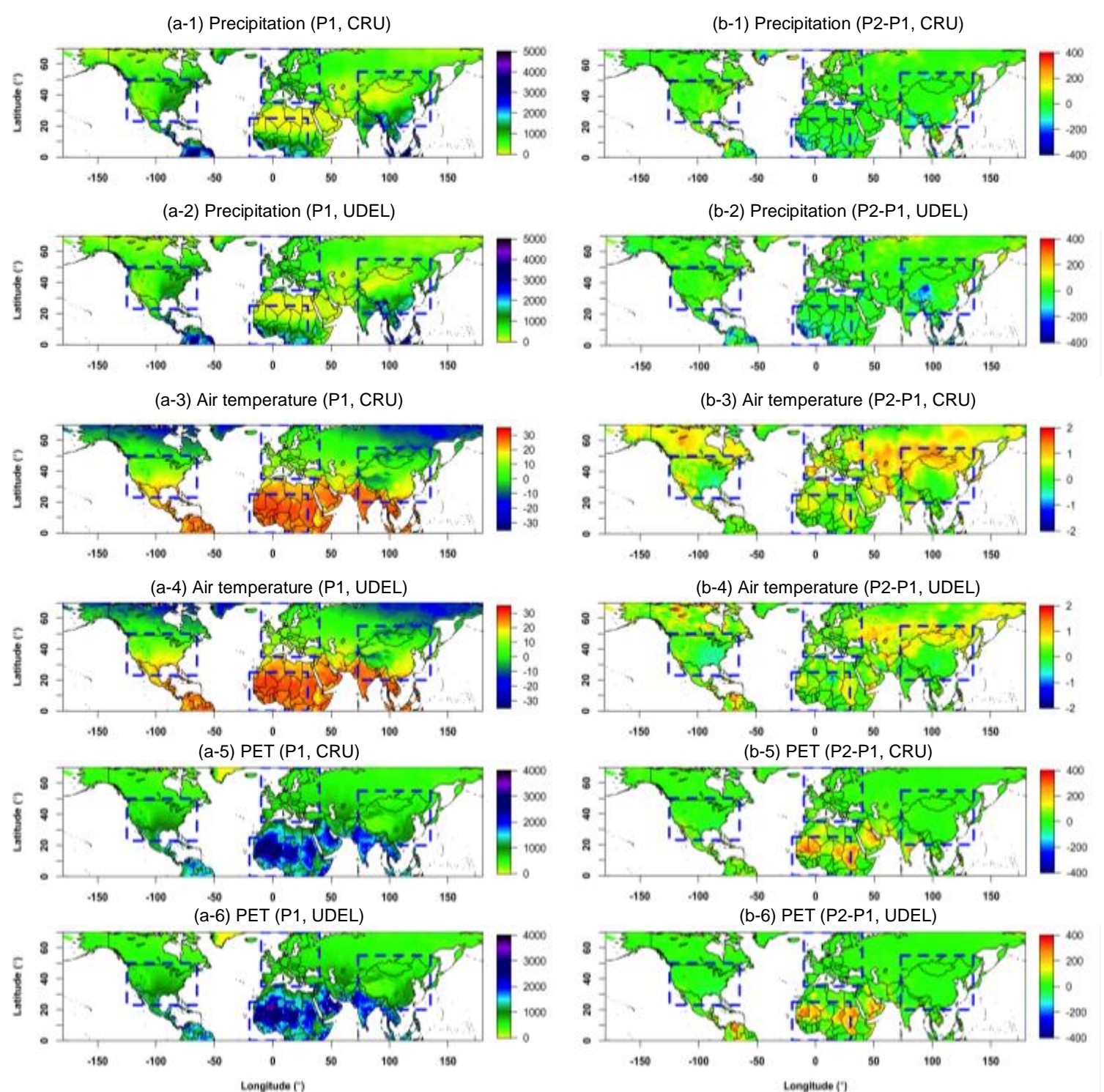

**Figure 2. Annual precipitation (mm), annual average air temperature (°C) and annual potential evapotranspiration (PET) (mm) based on the datasets of the Climate Research Unit (CRU) and the University of Delaware (UDEL) for the period of 1901-1957 (P1) and the difference between P1 and 1958-2014 (P2) (i.e., P2-P1).**

(a) Annual precipitation

(b) Annual potential evapotranspiration

(c) Annual surplus or deficit

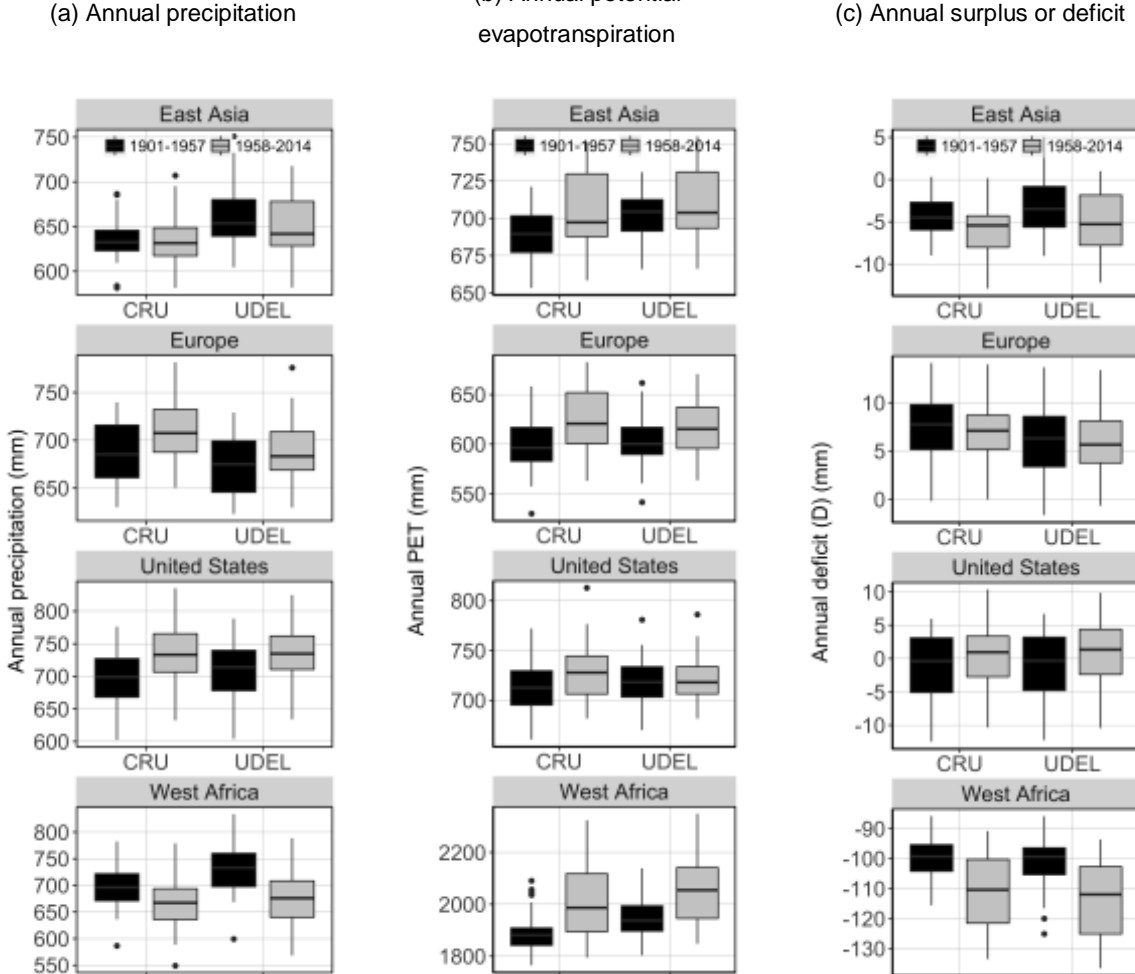

**Figure 3. Temporal variations in annual precipitation (mm), potential evapotranspiration (PET) (mm) and surplus or deficit (D) (mm) based on two datasets (CRU and UDEL) and periods (1901-1957 and 1958-2014). In the box plots, the center line represents the median value; the top and bottom of each box represent the 25th and 75th percentiles of the data, respectively; and the dots represent outliers.**

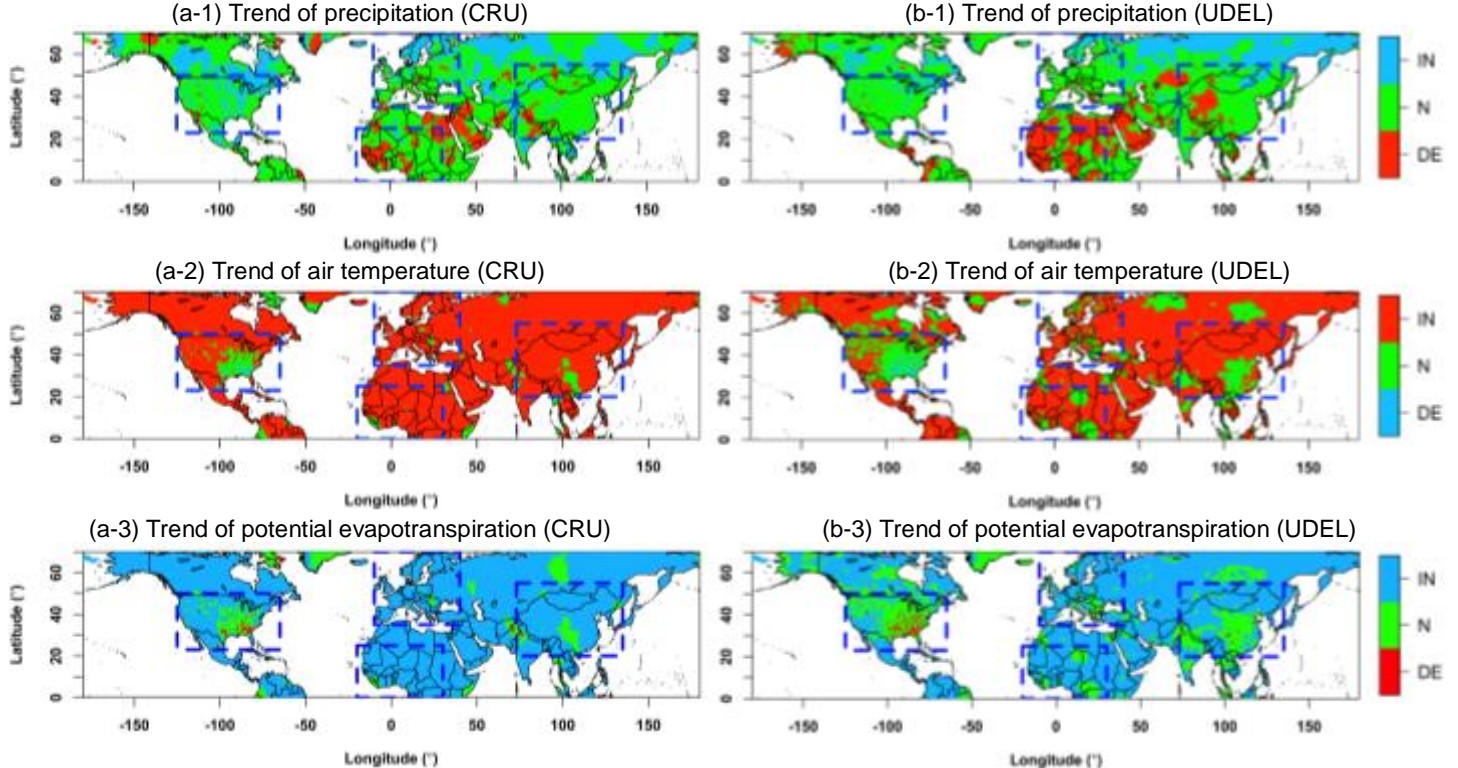

**Figure 4. Trends in annual precipitation, annual average temperature and annual potential evapotranspiration based on the CRU and UDEL datasets. The colored regions correspond to regions of IN, N and DE, which indicate a statistically positive (increasing) trend, no trend and a negative (decreasing) trend, respectively, at a significance level of 5%.**

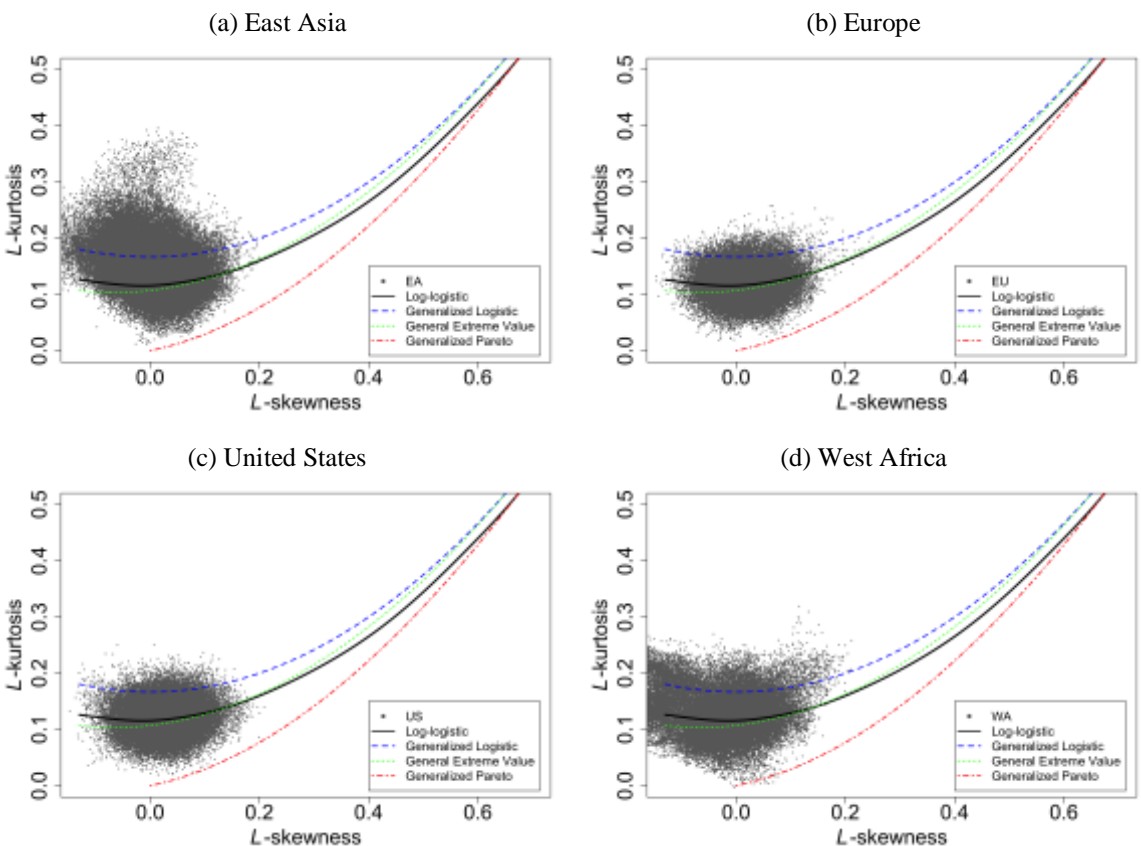

**Figure 5. L-moment ratio diagrams for the annual surplus or deficit (D) in Eq. (1) with a 12-month timescale based on CRU for each region from 1901-2014 and 1901-1957.**

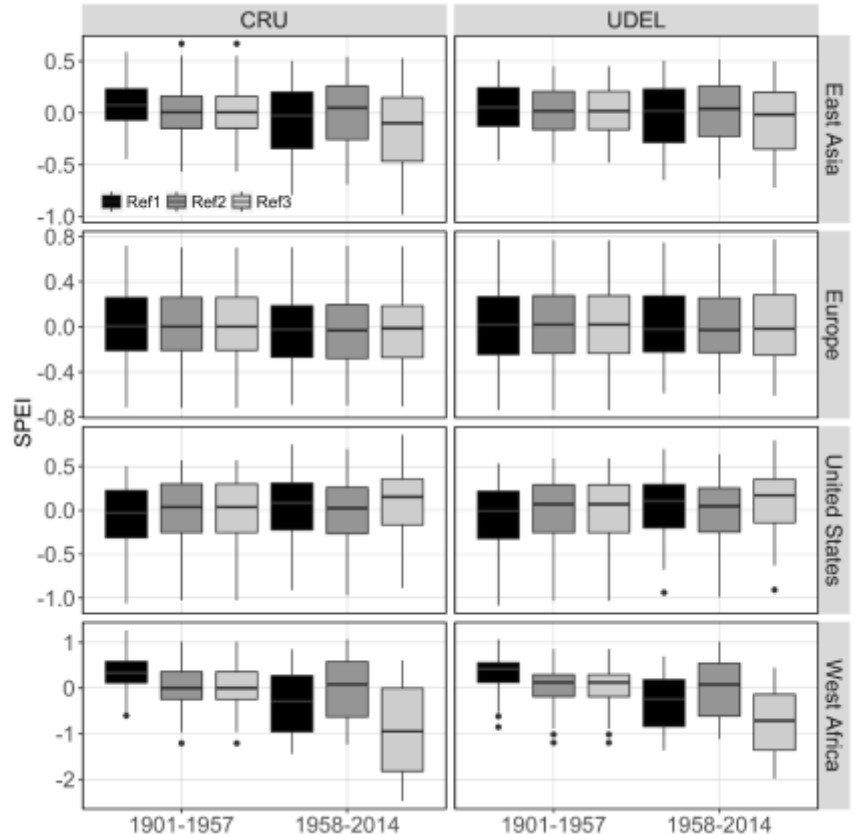

**Figure 6. Temporal variations in SPEI-12 for three different reference periods (Ref1, Ref2 and Ref3 in Table 2) based on the CRU and UDEL datasets from 1901–1957 and 1958–2014. In the box plots, the center line represents the median value; the top and bottom of each box represent the 25$^{th}$ and 75$^{th}$ percentiles of the data, respectively; and the dots represent outliers.**

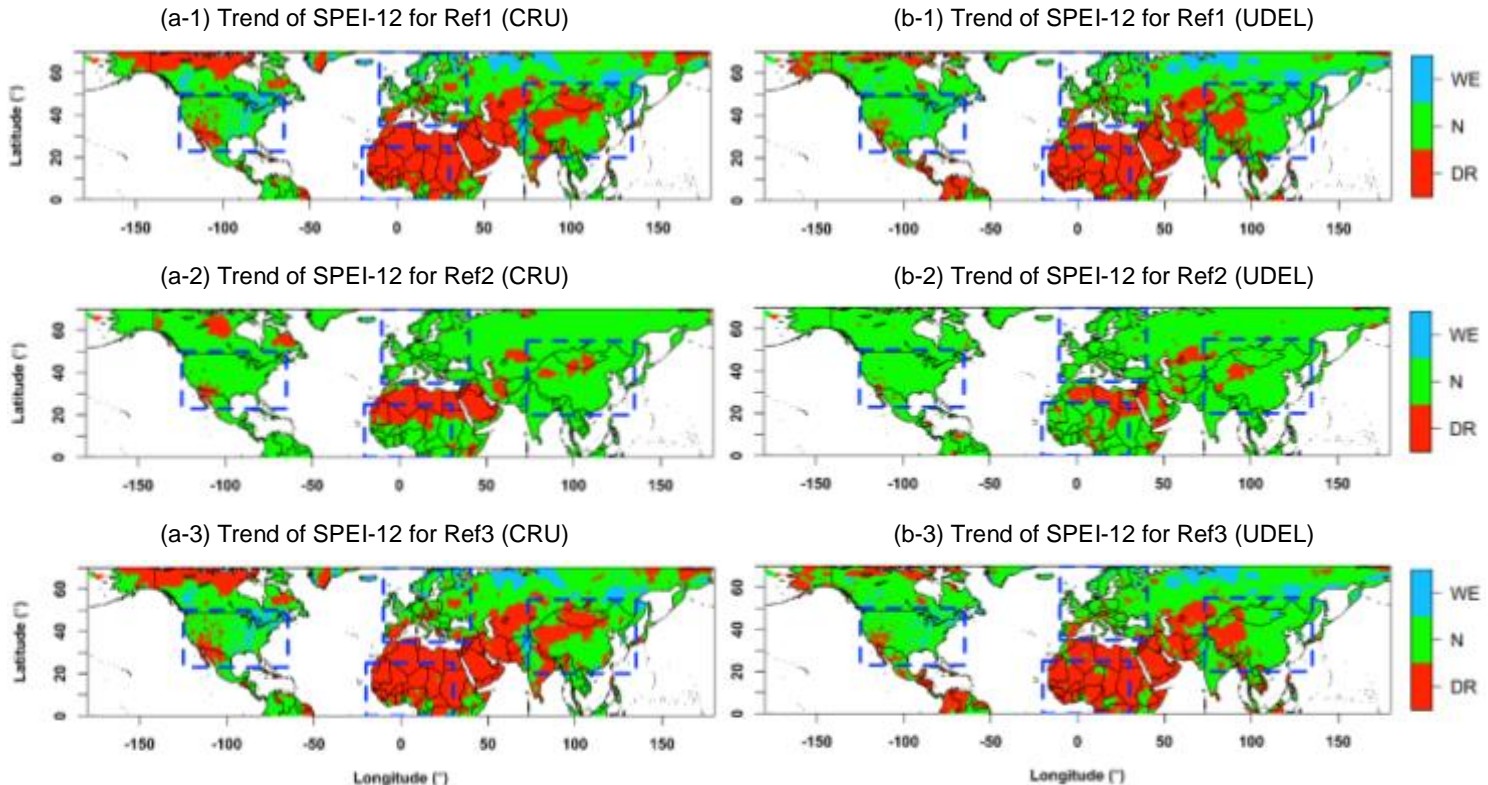

**Figure 7. SPEI-12 trends in three different reference periods (Ref1, Ref2 and Ref3 in Table 2) based on the (a) CRU and (b) UDEL datasets. The colored regions correspond to regions of WE, N and DR, which denote a statistically positive (wetting) trend, no trend and a negative (drying) trend, respectively, at a significance level of 5%.**

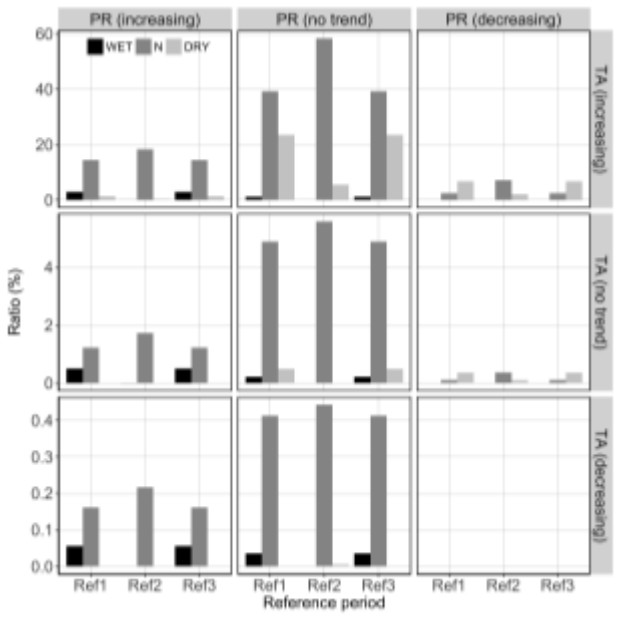

(a-1) Precipitation vs. Air temperature (CRU)

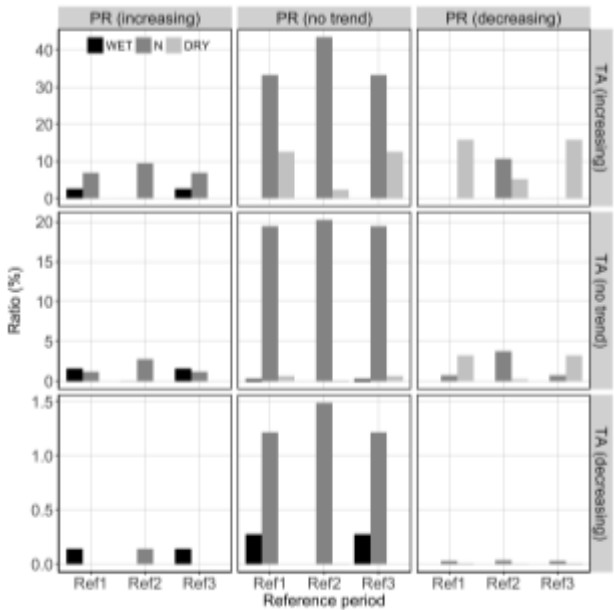

(b-1) Precipitation vs. Air temperature (UDEL)

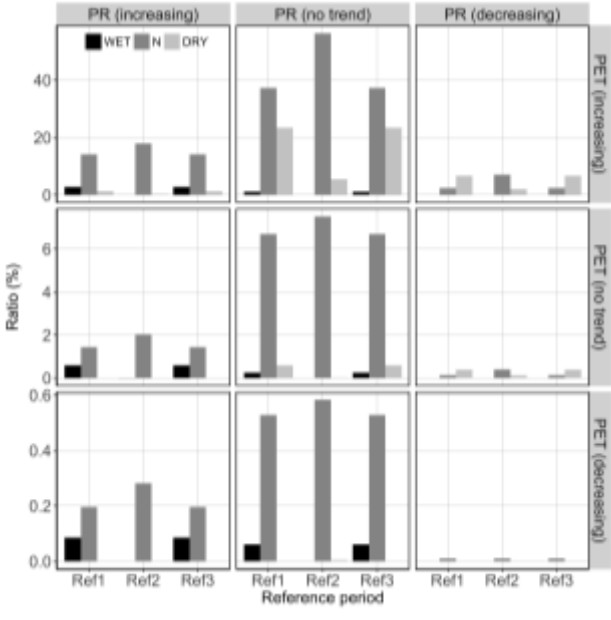

(a-2) Precipitation vs. Potential evapotranspiration (CRU)

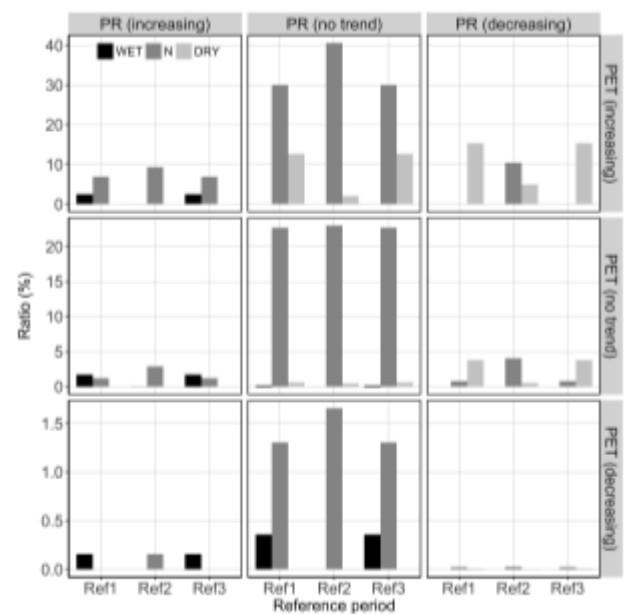

(b-2) Precipitation vs. Potential evapotranspiration (UDEL)

**Figure 8. The ratios of SPEI-12 trends in three different reference periods (Ref1 to Ref3 in Table 2) based on the CRU and UDEL datasets for trends of monthly precipitation and temperature (or potential evapotranspiration) in each region. For the SPEI, WET, N and DRY indicate a statistically positive (wetting) trend, no trend and a negative (drying) trend, respectively, at a significance level of 5%. Trends of precipitation, air temperature and potential evapotranspiration are grouped into three with a statistically increasing (positive) trend, no trend and a decreasing (negative) trend at a significance level of 5%.**

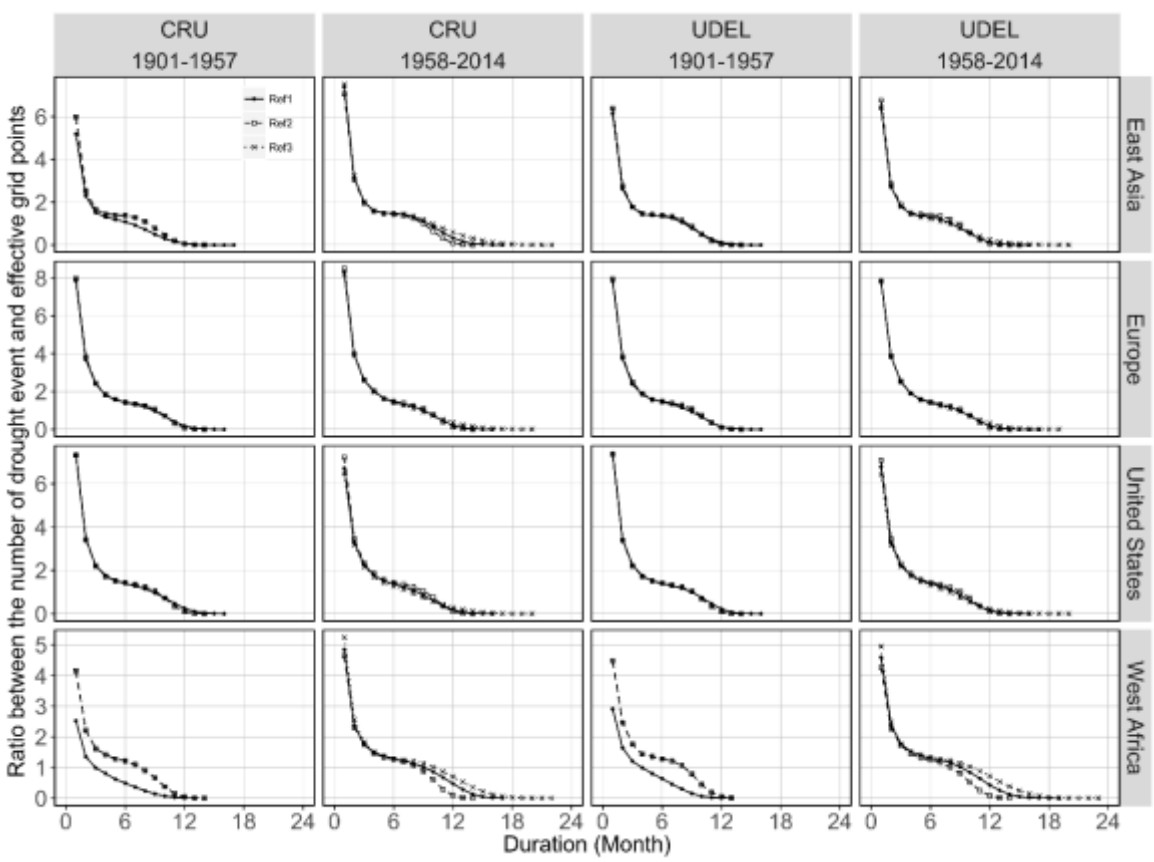

**Figure 9. Ratio between the number of drought events and the number of terrestrial data grid points in three different reference periods (Ref1, Ref2 and Ref3 in Table 2) based on the CRU and UDEL datasets from 1901–1957 and 1958–2014 in each region.**

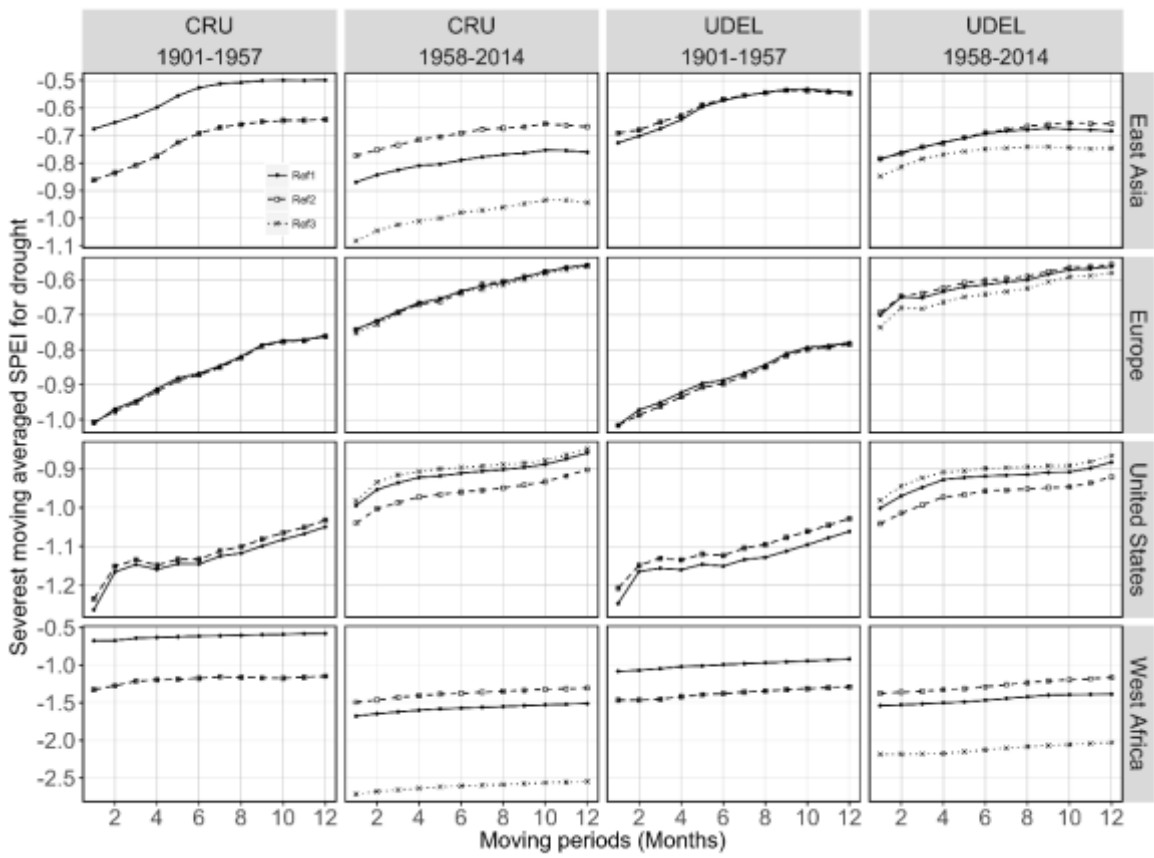

**Figure 10. Most severe moving average of regional SPEI-12 over 1–12 months for three different reference periods (Ref1, Ref2 and Ref3 in Table 2) based on the CRU and UDEL datasets from 1901–1957 and 1958–2014 in each region.**

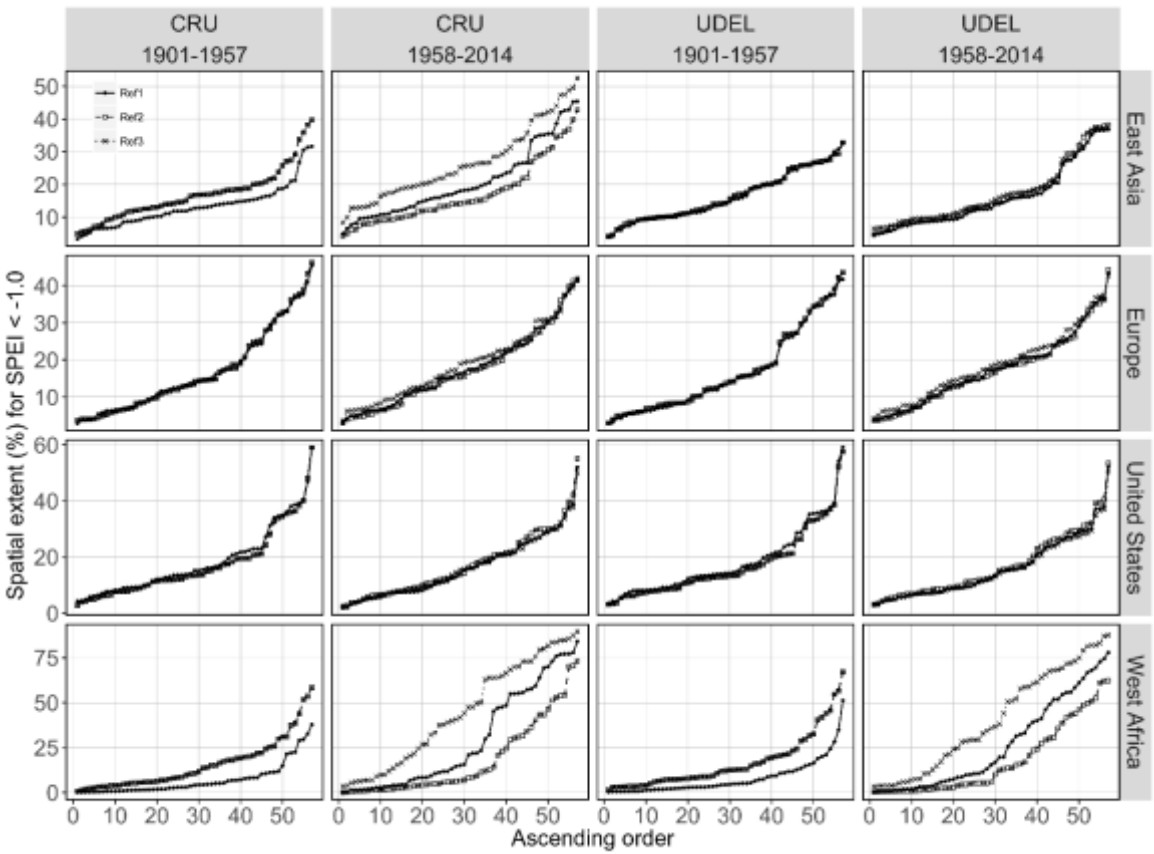

**Figure 11. Spatial extent (%) of SPEI-12 < -1.0 for three different reference periods (Ref1, Ref2 and Ref3 in Table 2) based on the CRU and UDEL datasets from 1901–1957 and 1958–2014 in each region.**

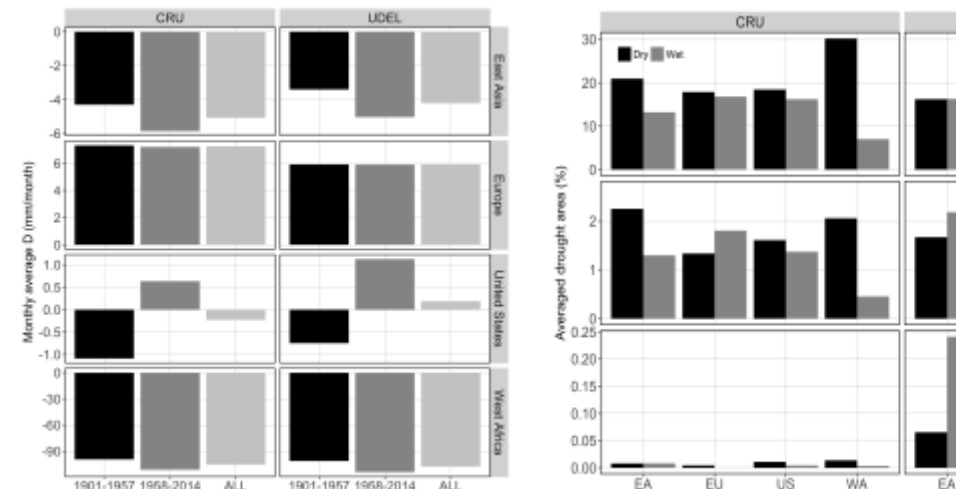

(a) Monthly average D                    (b) Averaged drought area (%)

**Figure 12. Monthly average surplus or deficit (D) (mm) in Eq. (1) and average drought area (%) based on the CRU and UDEL datasets in East Asia (EA), Europe (EU), the United States (US) and West Africa (WA) regions for the Ref1 condition. In (a), ALL denotes the period of 1901-2014. In (b), Dry denotes that the monthly average D in the assessment period is less than that in the reference period, and Wet denotes that the monthly average D in the assessment period is greater than that in the reference period based on the Ref1 condition.**

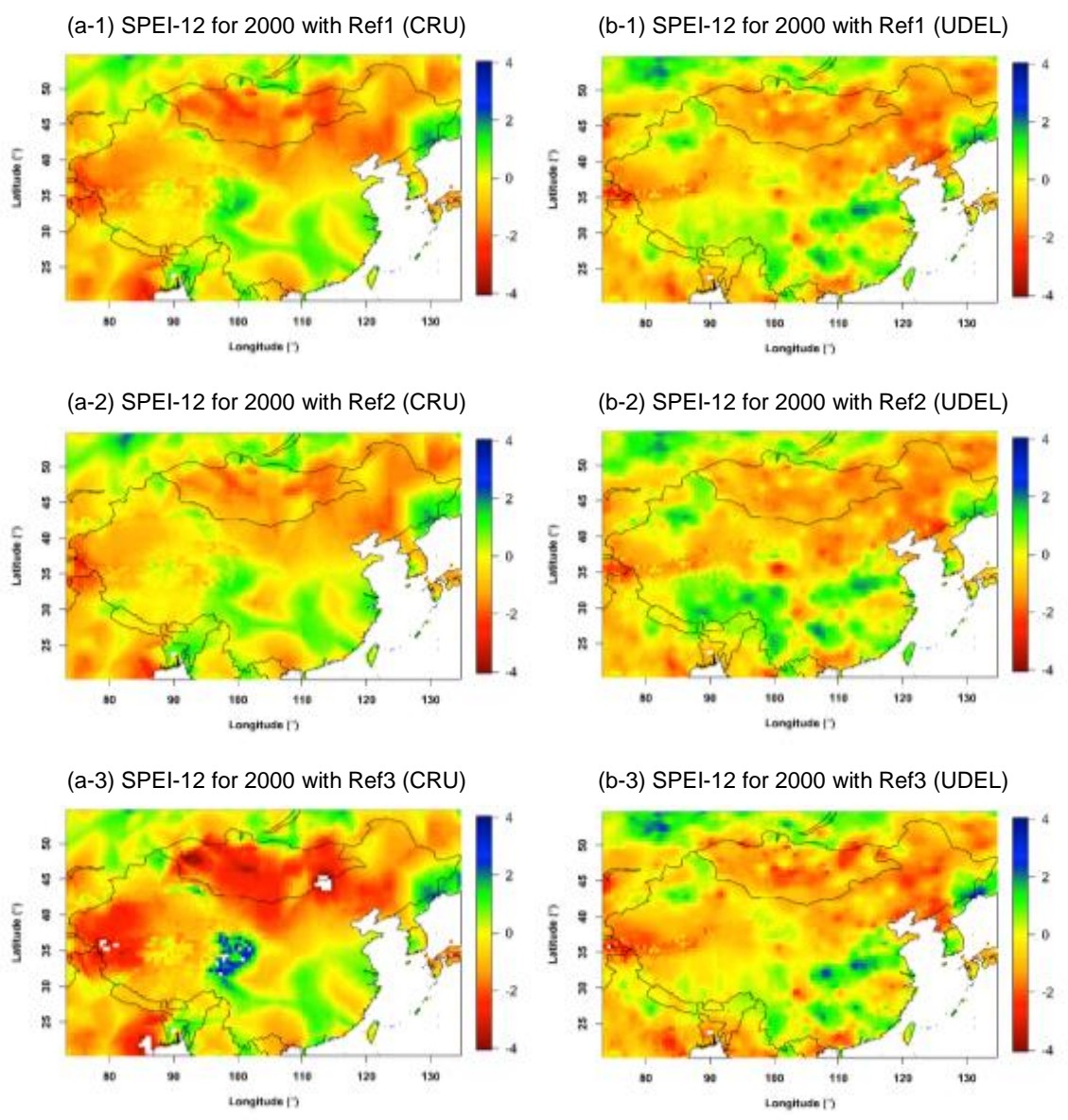

**Figure 13. Spatial distribution of SPEI-12 for three different reference periods (Ref1, Ref2 and Ref3 in Table 2) based on the (a) CRU and (b) UDEL datasets in East Asia in 2000 as an example.**

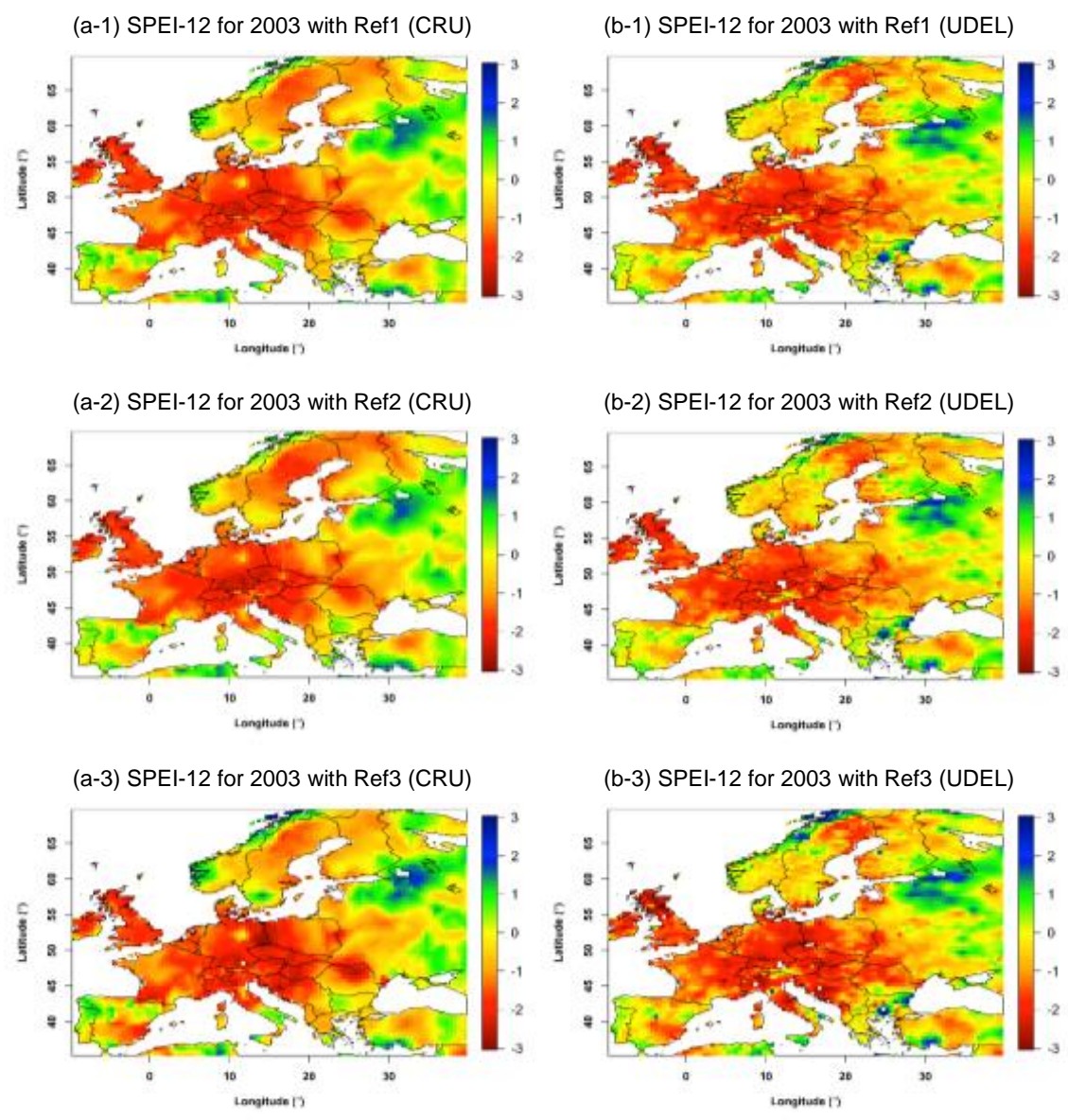

**Figure 14. Spatial distribution of SPEI-12 for three different reference periods (Ref1, Ref2 and Ref3 in Table 2) based on the (a) CRU and (b) UDEL datasets in Europe in 2003 as an example.**

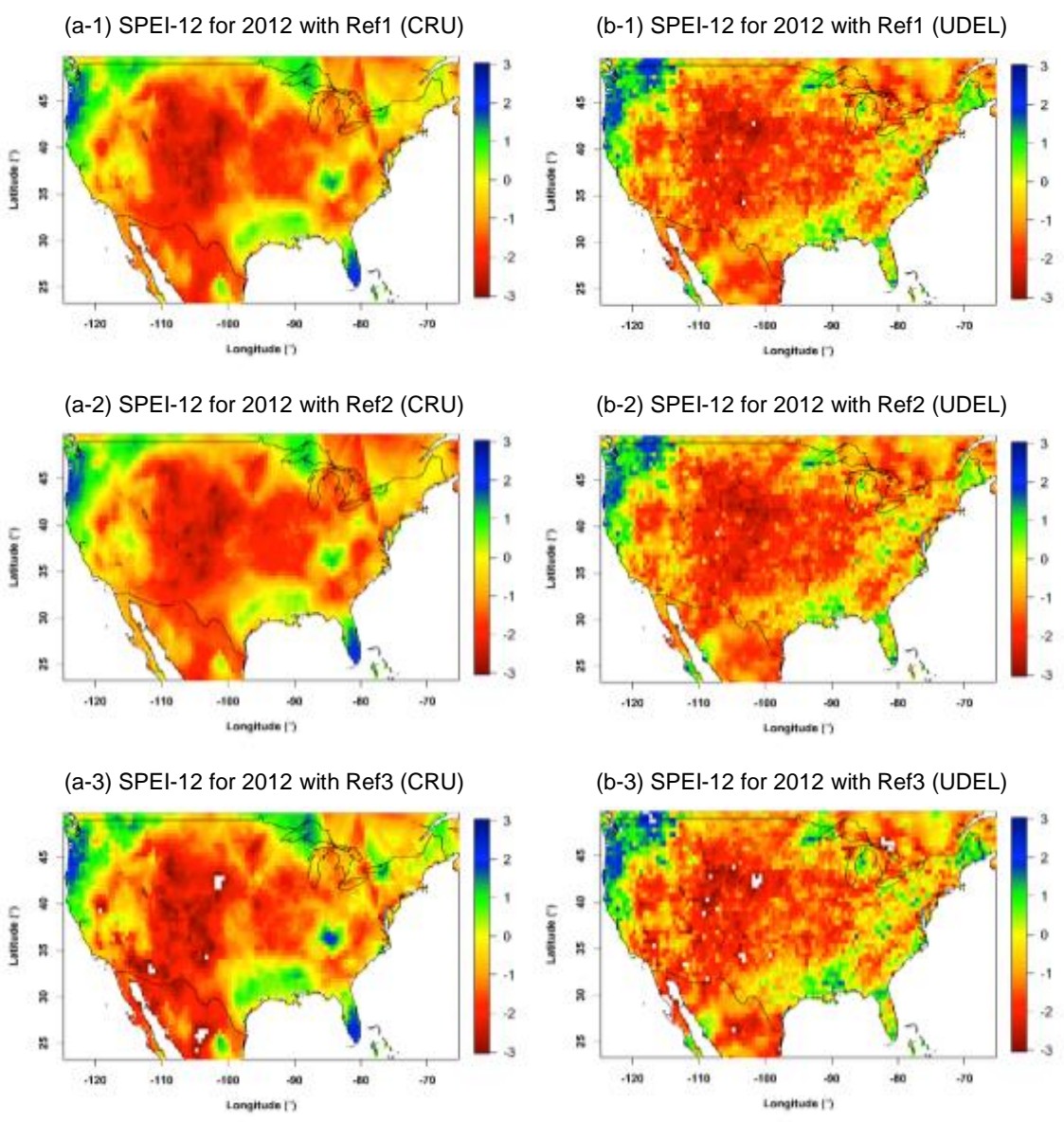

**Figure 15. Spatial distribution of SPEI-12 for three different reference periods (Ref1, Ref2 and Ref3 in Table 2) based on the (a) CRU and (b) UDEL datasets in the United States in 2012 as an example.**

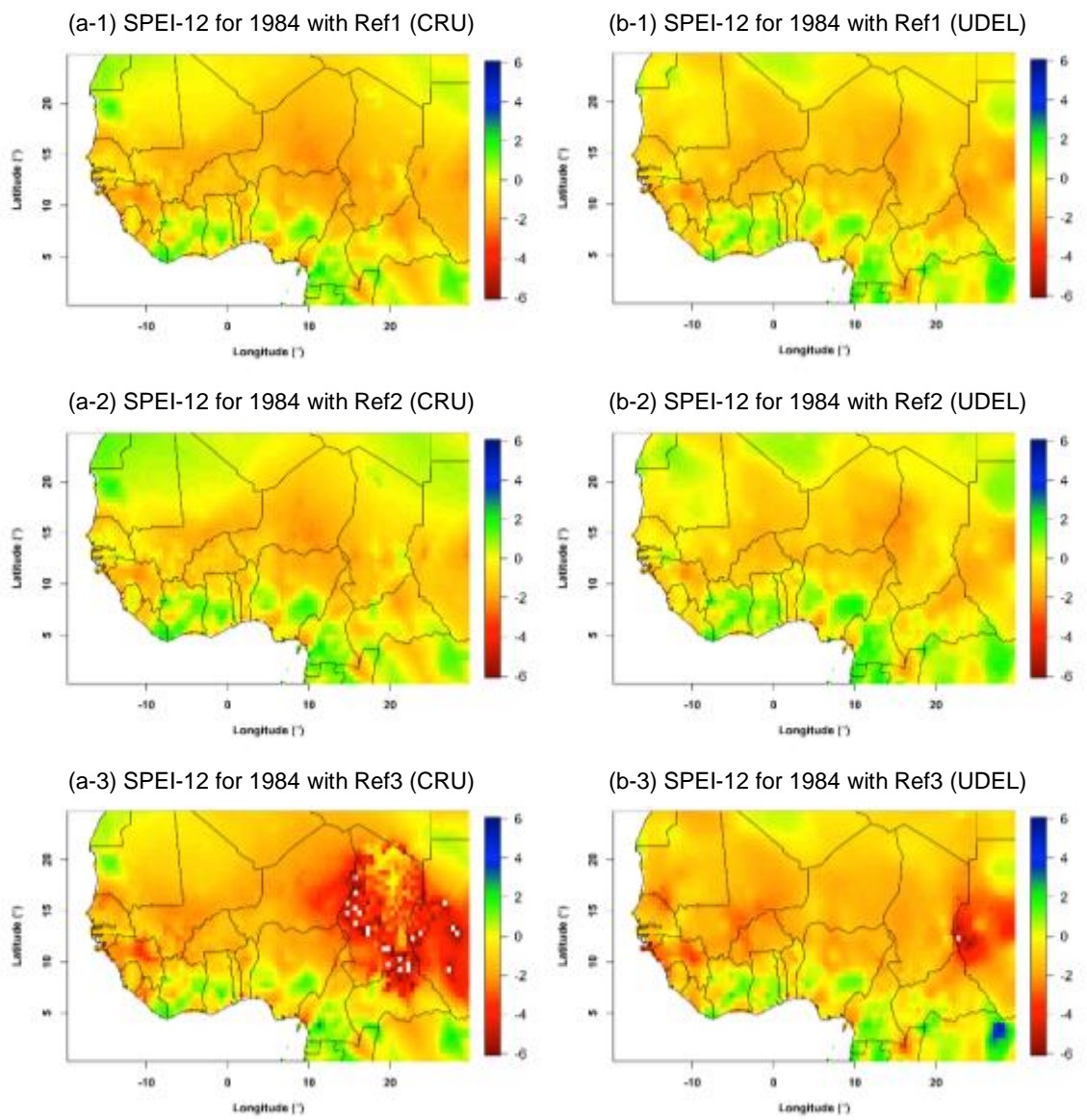

**Figure 16. Spatial distribution of SPEI-12 for three different reference periods (Ref1, Ref2 and Ref3 in Table 2) based on the (a) CRU and (b) UDEL datasets in West Africa in 1984 as an example.**

**Table 1. Classification of dry status in this study (modified from Mckee et al. (1993)).**

| Category | Description | SPEI |
|:---:|:---:|:---:|
| D1 | Moderately Dry or More Severe | $\leq$ -1.0 |
| D2 | Very Dry or More Severe | $\leq$ -2.0 |
| D3 | Extremely Dry or More Severe | $\leq$ -3.0 |

**Table 2. Definition of the cases of Ref1, Ref2 and Ref3 based on the estimation and calibration periods for SPEI-12.**

| Type | Estimation Period | Calibration Period |
|------|-------------------|--------------------|
| **Ref1** | 1901−1957 | 1901−2014 |
| | 1958−2014 | |
| **Ref2** | 1901−1957 | 1901−1957 |
| | 1958−2014 | 1958−2014 |
| **Ref3** | 1901−1957 | 1901−1957 |
| | 1958−2014 | |

**Table 3. Mean and standard deviation (STD) of precipitation and air temperature in each period and each region.**

| | | | CRU | | UDEL | |
|---|---|---|---|---|---|---|
| | | | 1901−1957 | 1958−2014 | 1901−1957 | 1958−2014 |
| Annual precipitation (mm) | East Asia | Mean | 637.19 | 635.52 | 659.67 | 649.21 |
| | | STD | 22.36 | 30.05 | 30.67 | 31.76 |
| | Europe | Mean | 685.86 | 711.03 | 674.17 | 688.31 |
| | | STD | 31.08 | 32.43 | 30.97 | 31.16 |
| | United States | Mean | 698.44 | 736.22 | 709.50 | 734.42 |
| | | STD | 43.31 | 41.48 | 44.06 | 41.55 |
| | West Africa | Mean | 698.49 | 666.59 | 734.84 | 676.11 |
| | | STD | 36.87 | 43.84 | 44.89 | 48.00 |
| Annual average air temperature (°C) | East Asia | Mean | 6.08 | 6.67 | 6.25 | 6.62 |
| | | STD | 0.28 | 0.52 | 0.31 | 0.48 |
| | Europe | Mean | 6.96 | 7.46 | 7.02 | 7.29 |
| | | STD | 0.56 | 0.68 | 0.55 | 0.64 |
| | United States | Mean | 10.46 | 10.78 | 10.59 | 10.64 |
| | | STD | 0.45 | 0.50 | 0.43 | 0.43 |
| | West Africa | Mean | 26.27 | 26.62 | 26.40 | 26.66 |
| | | STD | 0.25 | 0.48 | 0.26 | 0.41 |
| Annual potential evapotranspiration (mm) | East Asia | Mean | 688.69 | 705.78 | 700.44 | 709.52 |
| | | STD | 16.06 | 23.92 | 15.65 | 22.73 |
| | Europe | Mean | 598.06 | 624.48 | 603.15 | 617.35 |
| | | STD | 24.86 | 31.66 | 23.46 | 28.54 |
| | United States | Mean | 711.49 | 728.60 | 718.48 | 720.90 |
| | | STD | 23.84 | 26.28 | 22.59 | 21.28 |
| | West Africa | Mean | 1889.57 | 2001.37 | 1948.91 | 2044.28 |
| | | STD | 72.61 | 136.96 | 78.17 | 129.94 |

**Table 4. Spatial extent (%) (number of grid points relative to the total number of terrestrial grid points) in each region of different trends of SPEI-12 values based on different reference periods.**

| Zone | CRU | | | | | | UDEL | | | | | |
| --- | --- | --- | --- | --- | --- | --- | --- | --- | --- | --- | --- | --- |
| | Ref1 | | Ref2 | | Ref3 | | Ref1 | | Ref2 | | Ref3 | |
| | Wet | Dry | Wet | Dry | Wet | Dry | Wet | Dry | Wet | Dry | Wet | Dry |
| **East Asia** | 2.5 | 36.3 | 0.0 | 8.0 | 2.5 | 36.5 | 3.4 | 23.2 | 0.0 | 7.7 | 3.4 | 23.2 |
| Europe | 10.4 | 24.9 | 0.0 | 1.7 | 10.4 | 24.9 | 5.3 | 15.8 | 0.0 | 2.2 | 5.3 | 15.8 |
| United States | 18.6 | 16.2 | 0.0 | 67 | 18.6 | 16.2 | 11.3 | 9.7 | 0.1 | 3.1 | 11.3 | 9.7 |
| West Africa | 0.0 | 90.2 | 0.0 | 40.4 | 0.1 | 89.8 | 0.0 | 90.9 | 0.0 | 19.5 | 0.0 | 90.9 |

**Table 5. Monthly average D (mm/month) in each region for the periods of 1901-1957 (P1), 1958-2014 (P2) and 1901-2014 (P1+P2) based on the CRU and UDEL datasets.**

| | CRU | | | UDEL | | |
|---|---|---|---|---|---|---|
| | P1 | P2 | P1+P2 | P1 | P2 | P1+P2 |
| East Asia | -4.29 | -5.85 | -5.07 | -3.40 | -5.03 | -4.21 |
| Europe | 7.32 | 7.21 | 7.26 | 5.92 | 5.91 | 5.92 |
| United States | -1.09 | 0.64 | -0.23 | -0.75 | 1.13 | 0.19 |
| West Africa | -99.26 | -111.23 | -105.24 | -101.17 | -114.01 | -107.59 |

**Table 6. Spatial extent (%) (number of grid points in each drought category relative to the total number of grid points) of major drought events in different reference periods based on the CRU and UDEL datasets.**

| Zone | Period | Type | CRU | | | UDEL | | |
|---|---|---|---|---|---|---|---|---|
| | | | Ref1 | Ref2 | Ref3 | Ref1 | Ref2 | Ref3 |
| East Asia | 2000 | D1 | 32.63 | 27.48 | 38.80 | 26.81 | 27.39 | 29.62 |
| | | D2 | 2.45 | 0.75 | 14.64 | 0.92 | 0.73 | 2.64 |
| | | D3 | 0.05 | 0.00 | 1.83 | 0.04 | 0.01 | 0.07 |
| Europe | 2003 | D1 | 37.58 | 39.10 | 36.68 | 35.30 | 34.67 | 36.61 |
| | | D2 | 5.33 | 3.97 | 7.68 | 5.93 | 4.82 | 8.50 |
| | | D3 | 0.00 | 0.00 | 0.02 | 0.02 | 0.10 | 0.22 |
| United States | 2012 | D1 | 52.16 | 55.01 | 50.02 | 54.69 | 56.32 | 52.92 |
| | | D2 | 11.97 | 11.90 | 15.74 | 10.36 | 11.76 | 11.63 |
| | | D3 | 0.02 | 0.00 | 0.53 | 0.09 | 0.05 | 0.87 |
| West Africa | 1984 | D1 | 44.06 | 31.04 | 62.18 | 37.13 | 27.15 | 57.78 |
| | | D2 | 3.42 | 1.87 | 28.62 | 2.07 | 1.72 | 13.80 |
| | | D3 | 0.00 | 0.00 | 14.30 | 0.00 | 0.00 | 2.99 |