# Peer review of "Effects of different reference periods on drought index estimations for 1901-2014"

_Hydrology and Earth System Sciences, 2016_

## Referee Comment (RC1) · Anonymous Referee #1 · 3 Oct 2016

The impact of selected reference period on drought characteristics quantification is investigated in this paper. This is an interesting research question since the reference period matters under climate change. My major comment is that the research question is diluted many other factors (complexity) in the paper, e.g., two data sets (CRU and UDEL) and different regions. I encourage the authors to focus on the main research question (i.e., the impact of selected reference period on drought characteristics) and provide more insights on the reference period selection. For example, if climate is stationary, does the selected reference period matter? If the reference period is dry, are the quantified drought characteristics more severe in the assessment period? The authors may provide some suggestions/comments on the selection of reference period in practice? For example, is the early period or entire period of historical data recommend for the reference period? Or the author may provide a guideline to select reference period, e.g., the first step is to quantify the existence of trend in climate variables etc. The number of figures can be reduced. Line 26 page 1: change "hazard" to "hazards"? Line 31 on page 1: "state" to "stated"? Line 32 on page 1: "witnessed" to "have been witnessed"? Line 16 on page 2: rephrase the sentence: "They found that . . . . . . local regions" Line 35 on page 3 – Line 7 on page 4: The index and equations are difficult to follow due to the confusing indexes. For example, i is used for month index in line 35 on page 3; but it is used for year index in line 3 on page 4. Line 12 on page 4: "it one of" to "it is one of" Lines 24-29 on page 4: please correct the indexes for equations, e.g., equation (1) on page 4 should be equation (5) Lines 10 on page 5: change "relative to" to "and" Line 34 on page 8 – line 2 on page 9: Are these finding specific to this study or general to any regions and time periods?

---

## Author Comment (AC1) · 27 Oct 2016

**Response to Comments by Anonymous Reviewer #1**

The impact of selected reference period on drought characteristics quantification is investigated in this paper. This is an interesting research question since the reference period matters under climate change. My major comment is that the research question is diluted many other factors (complexity) in the paper, e.g., two data sets (CRU and UDEL) and different regions. I encourage the authors to focus on the main research question (i.e., the impact of selected reference period on drought characteristics) and provide more insights on the reference period selection. For example, if climate is stationary, does the selected reference period matter? If the reference period is dry, are the quantified drought characteristics more severe in the assessment period?

>> As suggested by the reviewer, we have performed additional analyses to provide further insight regarding the reference period selection while still providing the results from the multiple datasets and regions.

(1) We investigated how the precipitation and temperature trends are related to the SPEI trend based on the reference period to improve the understanding of the role of the selected reference period in a stationary climate.

Based on the grid-level trend analyses of precipitation, air temperature and SPEI-12, we categorize each grid cell based on increasing, decreasing or neutral trends for each variable (i.e., precipitation, air temperature and SPEI-12). For SPEI-12, increasing and decreasing trends represent wetting and drying trends. For 27 (3\*3\*3) different cases, we present the ratio of each case relative to the total number of cases (i.e., total number of effective grid cells in all four regions), as shown in the figure below.

First, the SPEI-12 trends are the same between Ref1 and Ref3, as the estimation periods share the one reference period in both Ref1 and Ref3 while each estimation period uses its own reference period in Ref2. Thus, the values of SPEI-12 are different in both cases, but the trends (i.e., relative values) are the same. Second, precipitation and air temperature exhibit neutral (or no) trends (in the center panel; presumably stationary climate), and the grid percentages of different trends in SPEI-12 vary between Ref1/Ref3 and Ref2. However, the ratio is relatively small, as most grid cells display increasing temperature trends. Finally, based on neutral precipitation and increasing air temperature trends in most grid cells, the numbers of cells with neutral and drying SPEI-12 trends are notably different between Ref1/Ref3 and Ref2. An increasing temperature trend can be observed in most regions; thus, it is important to consider its impact on SPEI.

We will add this figure and a related explanation to the revised manuscript at the end of section 3.2.

(b) UDEL

Figure. The SPEI trends with 12-month lag (SPEI12) for three different reference periods (Ref1 to Ref3) for the CRU and UDEL datasets based on the trends of monthly precipitation and temperature in the four zones

(2) To understand how the drought characteristics would change if the reference period is dry (the question the reviewer asked), we compare the drought area (%) for dry and wet cases in EA, EU, US and WA with defining dry and wet cases as below. We define the drought and wet cases with using a water surplus or deficit D (in Eq. (1), D=P-PET). We compare Ds between the reference period and estimation period. A value of D in the estimation period less than that in the reference period represents the dry case, i.e., the estimation period is drier than the reference period.

We perform such analyses only in Ref1 for estimation periods of 1901-1957 (P1) and 1958-2014 (P2) and a reference/calibration period from 1901-2014 (P1+P2). For dry and wet cases, we quantify the drought areas (%) according to the three different drought levels (D1, D2 and D3, which denote the cases of SPEI

---

## Referee Comment (RC2) · Anonymous Referee #2 · 18 Nov 2016

The authors use two global precipitation and temperature datasets to calculate the SPEI drought metric using various choices of the calibration period. By quantifying drought severity, duration and extent in maps and by aggregating SPEI values over four regions, conclusions are reached on 'best practice' in setting the calibration period. A very nice touch is that the authors have looked at specific record-dry years for the regions under consideration, and visualised the effects of the choice of calibration period on the drought estimate.

The SPEI, like many other drought metrics, is a standardised metric making its estimates for dry or wet conditions comparable over diverse climatological areas. The issue what calibration period to use, and the effects of not using the full-length of the period for which data is available as calibration, has been debated in the literature. This makes the study a very welcome contribution to the discussion.

[Figure]

However, my view on the manuscript that is currently submitted is that it may raise more questions than answers. There are quite a few things unclear and difficult to believe. Some of the choices made are unlucky, like the regions over which the SPEI is averaged. Some of these analyses need to be looked at again. Nevertheless, the analysis of the very dry years in the final parts of the paper show that the authors are capable of making some fine analysis - it is just a pity that they fail to observed some of the interesting aspects of their results.

The two main concerns relate 1) to the quality and validity of figs. 6, 7 and 8. I have trouble understanding what they mean and (especially fig. 7) can't be correct. 2) The analysis of figs. 9, 10, 11, 12, which touches at the essence of what the authors aim to investigate, is incomplete.

There are many other less serious concerns.

Main concerns

1. (a) fig. 6: I simply do not understand the quantity that is on the y-axis. The text (page 5, line 9-11) says: "the drought frequency as the ratio between the total number of drought events (...) relative to the total effective grid points." Are you calculating the number of grid points with SPEI12 $\leq$ -1, and then divide this number by the total number of grid points? Fig. 6 gives me ratios well above 1, so this can't be the case. There is also 'duration' on the x-axis. This is not the length of the time window over which the SPEI is calculated, is it? It seems to be the period for which a grid point stays at or below the SPEI12=-1 threshold, right?

   (b) Also fig. 7. This can't be true.There is a continuous upward line for Europe (and less so for the US) from 1901-1957 to 1958-2014. This would mean that in 1901 the moving average of SPEI was lowest for the coming century. I do not see dry years in this series like 1921, 1976 or 2003. All lines (for each period and region) have upward slopes. I think why this is (it is because

of the use of Thornthwaite) but this is not discussed anywhere. It is strange that fig. 7 has upward trends for EU and US, while none of this is seen in fig. 4

(c) fig. 8 is not understandable. The x-axis says 'ascending order', claiming that for 'ascending order' $\sim$ 60 for CRU 1958-2014 and West Africa, 100% has SPEI $\leq$ -1. What does this mean?

2. The analysis of figs. 9-12 is a good idea, but there are a few things the authors need to explain and re-do.

(a) it is not clear which SPEI12 value is taken. For instance, the 2003 heat wave in Europe, which coincided with a dry period, the height of the heat wave was early August 2003. The Spring was rather dry and the heat wave stopped when Autumn was a bit wetter than usual for France and Spain and in December, northern Europe received more rain than usual. The question is now: which month provides the SPEI12 value which is characteristic of the 2003 drought in Europe? SPEI12 is based on 12-month accumulated precipitation, so do you take the December value (so that the whole of 2003 is captured)? Or do you take the annual averaged SPEI12, which then has a small influence of January 2002 as well in it..... A pragmatic approach is to make time series of monthly SPEI12 values and then take the month in 2003 with the lowest value, but over which area to average? The whole of Europe is nonsense, since the heatwave was much more local than that.

(b) All figures 9-12 show that for Ref3, the SPEI values are off the scale for some areas. I think that this is the main issue with the Ref3 approach. The SPEI (and SPI) are more-or-less normally distributed. By using the calibration from one period (like 1901-1957), you run the risk that the metric 'explodes' beyond the range in which the SPEI/SPI lives when droughts occur in the 1958-2014 period which are (much) more severe than anything seen in the

calibration period. Essentially, using Ref3, you are not only assuming stationarity of the climate but also that the the whole probability distribution of droughts (and pluvials) is sampled in this period.

Other issues the authors may want to look into

1. The CRU dataset (and presumably the UDEL dataset as well) relax values to climatological values when data is insufficiently available. For Europe and North America, this will not be a big problem I think, but for West Africa and South Asia the number of records going back to 1901 are few and far in between. This means that the early period in these regions sees much less month-to-month variability as the more recent periods. Discuss the implications of this on your results.

2. The issue of the sensitivity has been raised earlier by Van der Schrier et al. 2013 (doi:10.1002-jgrd.50355) and Trenberth et al. 2014.

3. The Thornthwaite (1948) parameterization is directly related to temperature and has a huge trend. Even without a trend in precipitation amounts, the difference between the two will have a drying trend. This should be noted in the ms. and observed in the figures.

4. Sect. 2.2 It would be helpful for many readers what the descriptions are associated with the various SPI/SPEI values and the chance that 'severe' or 'extreme' drought is likely to occur. These are available in the McKee article or in: edo.jrc.ec.europa.eu - documents - factsheets - factsheet_spi.pdf

5. page 5, line 13-14. It is a good idea to see how large the region is with SPEI $\leq$ -1. However, simply counting grid squares does not work. You need to calculate area, where the grid areas are weighted with the cosine of the latitude.

6. page 6, lines 1-10. Interesting analysis, but the areas defined are more-or-less arbitrary. In Europe, for instance, there is a wetting trend in northern Europe and

a drying trend in southern Europe. It makes more sense to separate these two. Take a selection of the Giorgi regions: www.ipcc.ch - ipccreports - tar - wg1 - images - fig10-1.gif

7. page 6, Instead of calculating the trends in temperature, it makes more sense to calculate the Thornthwaite PET value. This makes this analysis directly comparable to the trends in precipitation.

8. fig. 2. Perhaps show the CRU climatology and the difference between UDEL and CRU? The pictures are very similar now.

9. fig. 4. What are we seeing? Is that the median, the 25th and 75th percentiles and min & max values? This is nowhere in the text or caption.

10. table 2. I see values of 26.1 degrees for area NA. This can't be North America (which is US in the text).

---

## Author Comment (AC2) · 4 Dec 2016

**Response letter to reviewer's comments**

Please find the responses to the reviewer's comments in the blue font and the revised manuscript.

**Anonymous Referee # 2**

The authors use two global precipitation and temperature datasets to calculate the SPEI drought metric using various choices of the calibration period. By quantifying drought severity, duration and extent in maps and by aggregating SPEI values over four regions, conclusions are reached on 'best practice' in setting the calibration period. A very nice touch is that the authors have looked at specific record-dry years for the regions under consideration, and visualised the effects of the choice of calibration period on the drought estimate.

The SPEI, like many other drought metrics, is a standardised metric making its estimates for dry or wet conditions comparable over diverse climatological areas. The issue what calibration period to use, and the effects of not using the full-length of the period for which data is available as calibration, has been debated in the literature. This makes the study a very welcome contribution to the discussion.

However, my view on the manuscript that is currently submitted is that it may raise more questions than answers. There are quite a few things unclear and difficult to believe. Some of the choices made are unlucky, like the regions over which the SPEI is averaged. Some of these analyses need to be looked at again. Nevertheless, the analysis of the very dry years in the final parts of the paper show that the authors are capable of making some fine analysis - it is just a pity that they fail to observed some of the interesting aspects of their results.

The two main concerns relate 1) to the quality and validity of figs. 6, 7 and 8. I have trouble understanding what they mean and (especially fig. 7) can't be correct. 2) The analysis of figs. 9, 10, 11, 12, which touches at the essence of what the authors aim to investigate, is incomplete.

There are many other less serious concerns.

Main concerns
1.
(a) fig. 6: I simply do not understand the quantity that is on the y-axis. The text (page 5, line 9-11) says: "the drought frequency as the ratio between the total number of drought events (...) relative to the total effective grid points." Are you calculating the number of grid points with SPEI12 $\leq$ -1, and then divide this number by the total number of grid points? Fig. 6 gives me ratios well above 1, so this can't be the case. There is also 'duration' on the x-axis. This is not the length of the time window over which the SPEI is calculated, is it? It seems to be the period for which a grid point stays at or below the SPEI12=-1 threshold, right?

>> First of all, the frequency can be well above 1. The multiple drought events can occur in one grid cell as the monthly SPEI-12 time series over the periods of 1901-195 and 1958-2014 are examined. Second, the duration is how long the SPEI-12 stays

at or below -1, not the length of the time window over the SPEI-12 is calculated. To clarify our approach to calculate the duration-frequency relation for Fig. 7 in the revised manuscript (Fig. 6 in the original manuscript), we have re-written the explanation about it.

Page 7: *"As explained above, a drought event is counted when the monthly SPEI-12 is estimated to be at or below -1.0 for the drought duration-frequency relation. For each drought event of grid cell, the duration is how long the SPEI-12 stays at or below -1. The frequency is the ratio between the total number of drought events and the effective grid points in each region (Fig. 7)."*

(b) Also fig. 7. This can't be true. There is a continuous upward line for Europe (and less so for the US) from 1901-1957 to 1958-2014. This would mean that in 1901 the moving average of SPEI was lowest for the coming century. I do not see dry years in this series like 1921, 1976 or 2003. All lines (for each period and region) have upward slopes. I think why this is (it is because of the use of Thornthwaite) but this is not discussed anywhere. It is strange that fig. 7 has upward trends for EU and US, while none of this is seen in fig. 4

>> Yes, we agree that the continuous upward line from P1 to P2 for EU and US seems strange. Therefore we have performed the analysis again and found the same results as in the original manuscript. To better understand our results in Fig. 9 of the revised manuscript (Fig. 7 of the original manuscript), we have performed an additional analysis as presented in Fig. 4 of the revised manuscript, presenting the averaged temporal variations of annual precipitation, PET and water surplus or deficit D (=P-PET in Eq.1). We have checked those variables as the PET increased significantly from P1 to P2 as the air temperature, which increasing significantly from P1 to P2, is a key controlling factor for the PET, as estimated based on the Thornthwaite approach in this study.

We therefore argue that such continuous upward line from P1 to P2 could be obtained because of the following reasons. First of all, we examine the regionally averaged indices, which do not necessarily capture the local severe drought events. Second, we find that it is consistent with Fig. 4. In US, the increase in precipitation is higher than that in PET, which leads the increase in D. In EU, the increase in PET is higher than that in precipitation, and thus the decrease in D is found in terms of average but the slight increase in the lower extreme of D is found. Therefore the severest drought events present less significant in P2 compared to those in P1. This point has been added in the manuscript as it follows.

Page 7: *"Such findings are seemingly inconsistent with the recently observed severe drought events in US and EU, but it is possible since we examine the regionally averaged indices, not the local extremes of SPEIs. Also it is consistent with Fig. 4. In US, the increase in precipitation is higher than that in PET, which leads the increase*

*in D (=P-PET in Eq.1). In EU, the increase in PET is higher than that in precipitation, and thus the decrease in D is found in terms of average but the slight increase in the lower extreme of D is found. Therefore the severest drought events present less severe in P2 compared to those in P1."*

[Figure]

**Figure 4. Temporal variations of annual precipitation, PET and surplus or deficit (D=P-PET) depending on two datasets (CRU and UDEL) and periods (1901-1957 and 1958-2014). In the box plot, the center line represents the median value; the top and bottom of box represent the 25th and 75th percentile of the data, respectively; the dot represents the outlier.**

(c) fig. 8 is not understandable. The x-axis says 'ascending order', claiming that for 'ascending order' ~ 60 for CRU 1958-2014 and West Africa, 100% has SPEI ≤ -1. What does this mean?

>> As per reviewer's suggestion, we have added the detailed explanation about how we derive Figure 9 in the revised manuscript (Fig. 8 in the original manuscript) as it follows.

Page 8: *"We count the numbers of grid points with the SPEI-12 less than -1.0 for each period (i.e., P1 and P2) and divide them with the effective grid numbers in the region to derive the spatial extent, i.e., the grid percentage of droughts. Then the annual time series of spatial extent are sorted in ascending order, from the smallest to the largest."*

>> In addition, the spatial extents in WA are as large as 89.6% with CRU and 87.7 % with UDEL for 1958-2014. It means that at least one year showed the drought events (with the SPEI-12 less than -1) occur over the 89.6% (87.7%) of the study area in WA with CRU (UDEL).

2. The analysis of figs. 9-12 is a good idea, but there are a few things the authors need to explain and re-do.
(a) it is not clear which SPEI12 value is taken. For instance, the 2003 heat wave in Europe, which coincided with a dry period, the height of the heat wave was early August 2003. The Spring was rather dry and the heat wave stopped when Autumn was a bit wetter than usual for France and Spain and in December, northern Europe received more rain than usual. The question is now: which month provides the SPEI12 value which is characteristic of the 2003 drought in Europe? SPEI12 is based on 12-month accumulated precipitation, so do you take the December value (so that the whole of 2003 is captured)? Or do you take the annual averaged SPEI12, which then has a small influence of January 2002 as well in it..... A pragmatic approach is to make time series of monthly SPEI12 values and then take the month in 2003 with the lowest value, but over which area to average? The whole of Europe is nonsense, since the heatwave was much more local than that.

>> We have examined the annual SPEI-12 based on the data from January to December in each year. Therefore our results show the average stage of the European heat wave in 2003. It is possible to take the suggested approach by the reviewer (i.e., taking the lowest value of monthly SPEI-12 for each grid cell), but it is not relevant to the focus of this study. Therefore it could be performed in the future study. In the revised manuscript, we have clarified our approach to estimate the SPEI-12 for Figs. 11-14 (Figs. 9-12 in the original manuscript) as well as possibly different results according to the local area as it follows.

Page 9: *"Here the annual SPEI-12s with the monthly climate data from January to December in each year are first constructed and then the SPEI-12s for a chosen year are examined in detail."*

Page 10: *"Although this study with historical data may shows the different results depending on the selected local area, a similar study with historical data or climate change scenarios in different regions would undoubtedly strengthen our findings."*

(b) All figures 9-12 show that for Ref3, the SPEI values are off the scale for some areas. I think that this is the main issue with the Ref3 approach. The SPEI (and SPI) are more-or-less normally distributed. By using the calibration from one period (like 1901-1957), you run the risk that the metric 'explodes' beyond the range in which the SPEI/SPI lives when droughts occur in the 1958-2014 period which are (much) more severe than anything seen in the calibration period. Essentially, using Ref3, you are

not only assuming stationarity of the climate but also that the the whole probability distribution of droughts (and pluvials) is sampled in this period.

>> Yes, we agree. As per reviewer's suggestion, we have added a relevant discussion in the revised manuscript as it follows.

Page 9: *"In particular, the several extremes (i.e., out of the scale ranges in Figs. 11-14) of SPEI-12 in Ref3 cases highlight the importance of the reference period. By using the reference period of the certain past time (P1 in this study, i.e., Ref3), the drought events in the estimation period could be beyond the range in which the distribution is calibrated for the index. Essentially, using Ref3, it is assumed that not only the stationarity of the climate but also that the whole probability distribution of droughts is sampled in this period."*

Other issues the authors may want to look into
1. The CRU dataset (and presumably the UDEL dataset as well) relax values to climatological values when data is insufficiently available. For Europe and North America, this will not be a big problem I think, but for West Africa and South Asia the number of records going back to 1901 are few and far in between. This means that the early period in these regions sees much less month-to-month variability as the more recent periods. Discuss the implications of this on your results.

>> We have added the suggested discussion about the lack of ground-based observations in the early period in the revised manuscript as it follows:

Page 6: *"Furthermore, the variances of SPEI are relatively small for P1 compared to those for P2 in EA and WA while no noticeable differences in the variances are captured in EU and US. It may attribute to the lack of ground-based observations before 1950 (i.e., the most of P1) (Becker et al., 2013; Vittal et al., 2013; Nasrollahi et al., 2015) and such limit in data availability seems play a role in reducing the variance of SPEI for P1 in EA and WA."*

2. The issue of the sensitivity has been raised earlier by Van der Schrier et al. 2013 (doi:10.1002-jgrd.50355) and Trenberth et al. 2014.

>> The past studies about the sensitivity in regard to the reference period have been included as it follows:

Page 2: *"It has already been pointed out for the self-calibrated PDSI that trends towards more extreme conditions are amplified when the calibration period does not include the recent part of data, including the recent effects of climate change (Van der Schrier et al., 2013; Trenberth et al., 2013)."*

3. The Thornthwaite (1948) parameterization is directly related to temperature and has a huge trend. Even without a trend in precipitation amounts, the difference between the two will have a drying trend. This should be noted in the ms. and observed in the figures.

>> We have pointed out the use of Thornthwaite and its influences throughout the revised manuscript as it follows.

Page 5: "Precipitation, air temperature and PET are investigated because they are used to estimate the SPEIs (Figs. 2 and 3 and Table 3). As noted already, the SPEIs are estimated based on the distribution of D (=P-PET in Eq. 1) and here the air temperature is directly related to PET because we use the Thornthwaite approach to estimate PET."

>> Furthermore, we have added the analyses of PET and D(=P-PET) in addition to the analyses of precipitation and air temperature in Figs. 3, 4 and 7 as well as Table 3 in the revised manuscript.

4. Sect. 2.2 It would be helpful for many readers what the descriptions are associated with the various SPI/SPEI values and the chance that 'severe' or 'extreme' drought is likely to occur. These are available in the McKee article or in: edo.jrc.ec.europa.eu - documents - factsheets - factsheet_spi.pdf

>> As per reviewer's suggestion, we have added Table 1 to show the categories of drought.

*Table 1. Classification of dry status in this study (Mckee et al., 1993).*

| Category | Description | SPEI |
|----------|-------------|------|
| D1 | Moderate dry | $\leq$ -1.0 |
| D2 | Extreme dry | $\leq$ -2.0 |
| D3 | Very extreme dry | $\leq$ -3.0 |

5. page 5, line 13-14. It is a good idea to see how large the region is with SPEI $\leq$ -1. However, simply counting grid squares does not work. You need to calculate area, where the grid areas are weighted with the cosine of the latitude.

>> We agree that the area of each grid cell varies significantly according to the latitude. Therefore we have revised the relevant text to point out that for the spatial extent of drought, we calculate the number of drought grid points relative to the total effective grid points in each region, not the drought area relative to the total area in each region. Consequently, we revised the definition word from "area" to "spatial extent" not to misunderstand and added the detail meaning in the manuscripts as it follows:

Page 8: *"The spatial extents of droughts for the annual SPEI-12 $\leq$ -1.0 are examined by sorting the results in ascending order (Fig. 10). We count the numbers of grid points with the SPEI-12 less than -1.0 for each period (i.e., P1 and P2) and divide them with the effective grid numbers in the region to derive the spatial extent, i.e., the grid percentage of droughts. Then the annual time series of spatial extent are sorted in ascending order, from the smallest to the largest."*

6. page 6, lines 1-10. Interesting analysis, but the areas defined are more-or-less arbitrary. In Europe, for instance, there is a wetting trend in northern Europe and a drying trend in southern Europe. It makes more sense to separate these two. Take a selection of the Giorgi regions: www.ipcc.ch - ipccreports - tar - wg1 - images - fig10-1.gif

>> Yes, our selection of study region is more-or-less arbitrary. Each region could be divided into the sub-regions based on the climate classification as the reviewer suggested. While different studies use different climate sub-regions in their analyses, we more focus on the detailed changes within the large region such as EA, WA, US and EU. In this study, Fig. 4 and Table 3 do not show the detailed change in precipitation, air temperature and PET with the areal averages, but Figs 2 and 3 show the spatially distributed maps of averages and temporal changes. With putting Figs 2, 3 and 4 and Table 3 together, we were able to present different trends within the region in this study. Indeed, Fig. 3 presents a wetting trend in the northern Europe and a drying trend in the southern Europe. Therefore we would perform such sub-regional analyses based on the climate classification in the future study as it is pointed out in the revised manuscript.

Page 3: *"We perform the analyses based on the spatially distributed patterns over those regions as well as their averages, but without distinguishing the sub-regions based on the climate characteristics."*

Page 10: *"Although this study with historical data may shows the different results depending on the selected local area, a similar study with historical data or climate change scenarios in different regions would undoubtedly strengthen our findings."*

7. page 6, Instead of calculating the trends in temperature, it makes more sense to calculate the Thornthwaite PET value. This makes this analysis directly comparable to the trends in precipitation.

>> As per reviewer's suggestion, we have added the analyses of PET throughout the manuscript. Figs. 3, 4 and 7, and Table 3 are revised to include PET in addition to the precipitation and air temperature.

[Figure]

**Figure 3. Trends of annual precipitation, annual averaged temperature and annual PET for the CRU and UDEL datasets. PR and TA denote precipitation and temperature, respectively, and IN, N and DE indicate increasing, no trend and decreasing, respectively.**

[Figure]

**Figure 4. Temporal variations of annual precipitation, PET and surplus or deficit (D=P-PET) depending on two datasets (CRU and UDEL) and periods (1901-1957 and 1958-2014). In the box plot, the center line represents the median value; the top and bottom of box represent the 25th and 75th percentile of the data, respectively; the dot represents the outlier.**

8. fig. 2. Perhaps show the CRU climatology and the difference between UDEL and CRU? The pictures are very similar now.

>> As we focus on the difference between P1 and P2, we have added the difference figure between P1 and P2 in Figure 2.

[Figure]

**Figure 2. Annual precipitation (mm), annual averaged temperature (°C) and annual PET (mm) for the CRU and UDEL datasets for P1 and their difference between P1 and P2.**

9. fig. 4. What are we seeing? Is that the median, the 25th and 75th percentiles and min & max values? This is nowhere in the text or caption.

>> As per reviewer's suggestion, we have added the detailed explanation for the box plot.

Figure 6 in the revised manuscript: *"In the box plot, the center line represents the median value; the top and bottom of box represent the 25th and 75th percentile of the data, respectively; the dot represents the outlier."*

10. table 2. I see values of 26.1 degrees for area NA. This can't be North America (which is US in the text).

>> We have corrected North America to US in abstract and Table 3.

Page 1: *"Focusing on East Asia, Europe, United States and West Africa"*

**Effects of different reference periods on drought index estimations for 1901-2014**

Myoung-Jin Um[1], Yeonjoo Kim[1,*], Daeryong Park[2], Jeongbin Kim[1]

[1]Department of Civil and Environmental Engineering, Yonsei University, Seoul, 03722, Republic of Korea
[2]Department of Civil, Environmental and Plant Engineering, Konkuk University, Seoul 05029, Republic of Korea

*Correspondence to*: Yeonjoo Kim (yeonjoo.kim@yonsei.ac.kr)

**Abstract.** This study aims to understand how different reference periods (i.e., calibration periods) of climate data for estimating the drought index influence regional drought assessments. Specifically, we investigate the influence of different reference periods on historical drought characteristics such as trends, frequency, intensity and spatial extents using the standard precipitation evapotranspiration index with a 12-month lag (SPEI-12) estimated from the datasets of the climate research unit (CRU) and the University of Delaware (UDEL). For the 1901–1957 (P1) and 1958–2014 (P2) estimation periods, three different types of reference periods are used: P1 and P2 together, P1 and P2 separately and P1 only. Focusing on East Asia, Europe, United States and West Africa, we find the influence of the reference periods to be significant in East Asia and West Africa, with dominant drying trends from P1 to P2. The reference periods influence the assessment of drought characteristics, particularly for severity and spatial extent, whereas their influence on the frequency is relatively small. Finally, self-calibration, which is the most common practice with an index such as SPEI, tends to underestimate the drought severity and spatial extent relative to the other approaches used in this study. Although the conclusions drawn in this study are limited to two global datasets, they nevertheless highlight the need for the reference period to be clarified in drought assessments to better understand regional drought characteristics and their temporal changes, particularly under climate change scenarios.

**1 Introduction**

Drought is a complex, slow-onset natural phenomenon affecting more people than any other hazards and seriously influencing water resources, agriculture, society and ecosystems (Hagman, 1984; Wilhite, 2002; Ionita et al., 2015). As drought impacts are largely nonstructural and spread over a relatively large region, the onset and end of a drought as well as its severity are often difficult to determine (Wilhite, 2002). Furthermore, based on recent changes in the 21st century and projected climate warming, such drought phenomena will likely worsen (Sheffield and Wood, 2008; Dai, 2010). Sheffield et al. (2012) stated that the severe and prolonged drought events have been witnessed since the 1970s and their changes are related to higher temperatures and lower precipitation.

[revised manuscript text omitted]

**2.2 Meteorological drought index**

Various drought indices have been used to understand different types of droughts, including meteorological drought, agricultural drought and hydrological drought (Heim, 2002). For meteorological droughts, the indices include the PDSI (Palmer, 1965), the SPI (McKee et al., 1993) and the SPEI (Vicente-Serrano et al., 2010). As different studies used different meteorological drought indices (Seneviratne, 2012; Sheffield et al., 2012; Trenberth et al., 2014; Nasrollahi et al., 2015; Touma et al., 2015), this study focuses on the SPEI. Devised by Vicente-Serrano et al. (2010), the SPEI has the advantage of being able to consider the effects of temperature variability for the drought relative to the SPI (Naumann et al., 2014) because the potential evapotranspiration (PET) can be calculated with air temperature based on Thornthwaite (1948). The SPEI uses the amount of precipitation minus PET and fits the data to the log-logistic probability distribution function. Here, we summarize the steps to estimate SPEI based on monthly precipitation and temperature. The detailed procedure for estimating the SPEI is well presented in Vicente-Serrano et al. (2010).

Step 1: Estimate the water surplus or deficit in month j ($D_j$) using the difference between precipitation ($P_j$) and potential evapotranspiration ($PET_j$):

$$D_j = P_j - PET_j \tag{1}$$

Here, the potential evapotranspiration is estimated based on the method of Thornthwaite (1948), which requires the monthly temperature, latitude, day and month.

Step 2: Estimate the accumulated difference ($X_{i,j}^k$) over the time scale $k$ in a given month $j$ and year $i$. For example, the accumulated difference for a month in a particular year with a 12-month time scale is calculated as follows:

$$X_{i,j}^k = \sum_{l=13-k+j}^{12} D_{i-1,l} + \sum_{l=1}^{j} D_{i,j}, \qquad if\ j < k \tag{2}$$

$$X_{i,j}^k = \sum_{l=j-k+1}^{j} D_{i,l}, \qquad if\ j \geq k \tag{3}$$

Step 3: Fit the accumulated difference to a log-logistic distribution as follows:

$$F(X) = \left[1 + \left(\frac{\alpha}{x-\gamma}\right)^\beta\right]^{-1} \tag{4}$$

where $F(X)$ is the cumulative probability function of a three-parameter log-logistic distribution with $\alpha$, $\beta$ and $\gamma$ representing the scale, shape and origin parameters, respectively. For the model fitting, the L-moment procedure (Hosking, 1990) is employed as it is one of the most robust and easy-to-use approaches.

Step 4: Estimate the SPEI based on the estimated $F(X)$. The SPEI can be derived from the standardized values of $F(X)$ and the classical approximation of Abramowitz and Stegun (1965) following Vicente-Serrano et al. (2010). The estimated drought index is classified as in Table 1 for moderate, extreme and very extreme dry cases. In this study, we focus on the SEPI with the 12-month lag (SPEI-12). SPEI can be estimated for different lag times such as 1, 3, 6, 9, 12 and 24 months.

**2.3 Temporal trends and statistical characteristics**

This study investigates various measures of historical droughts, including trend, frequency, severity and spatial extent (Lloyd-Hughes and Saunders, 2002; Wang et al., 2011; Hoerling et al., 2012; Seneviratne, 2012; Trenberth et al., 2014; Touma et al., 2015).

The temporal trend is investigated with a nonparametric and monotonic trend test with the S-statistic (Mann, 1945; and Kendall, 1976) as follows:

$$S = \sum_{i=1}^{n-1} \sum_{j=i+1}^{n} sgn(x_j - x_i) \tag{5}$$

$$where\ sgn(x_j - x_i) = \begin{cases} +1, & (x_j - x_i) > 0 \\ 0, & (x_j - x_i) = 0 \\ -1, & (x_j - x_i) < 0 \end{cases} \tag{6}$$

where $sgn$ is the sign function and $n$ is the sample size. The statistical significance of the trend can be predicted by a Z test as follows:

$$Z = \begin{cases} (S-1)/\sigma_s, & if\ S > 0 \\ 0, & if\ S = 0 \\ (S+1)/\sigma_s, & if\ S < 0 \end{cases} \tag{7}$$

$$\sigma_s = \sqrt{\left(n(n-1)(2n+5) - \sum_{j=1}^{q} t_j(t_j - 1)(2t_j + 5)\right)/18} \tag{8}$$

where $\sigma_s$ is the square root of S in the case that the $x$ values are possible tie situations, $q$ is the number of ties in the dataset and $t_j$ is the number of data in the $j$th tie group. The trend in the data does not exist for $Z < Z_{\alpha/2}$ at the significance level $\alpha$.

For the frequency, severity and spatial extent of drought, different measures have been defined and used in past studies (e.g., Wang et al., 2011; Touma et al., 2015; Um et al., 2016) because it is not straightforward to define these quantities in practice. For example, Touma et al. (2015) defined the duration, occurrence and spatial extent of drought to investigate the drought changes with 15 CMIP5 models throughout the world for the 21st century: the duration of drought is defined as the consecutive period below a certain drought status, the occurrence of droughts is defined as the total number of droughts in the period of interest, and the spatial extent of droughts is defined as the percentage of grid points below the given drought level, in which the corresponding drought index is less than the given drought category for each month.

In this study, we define three measures of droughts with the SPEI-12: (1) the drought frequency as the ratio between the total number of drought events, which is defined as the SPEI-12 ≤ -1, and the total effective grid points; (2) the severity as the lowest estimates among the regional monthly average SPEI-12 with moving windows with periods of 1 to 12 months; here, the regional averages are estimated for the four study regions depicted in Fig. 1; and (3) the spatial extent as the number of grids with the annual SPEI-12 ≤ -1.0 relative to the total grids.

**2.4 Design of data analysis**

To understand the influence of the reference period (i.e., calibration period) on the drought index, three different types of reference periods are used to estimate the SPEI-12 with the CRU and UDEL. To analyze separately the drought characteristics for the estimation periods of 1901–1957 (P1) and 1958–2014 (P2), different sets of reference periods are used (Table 2). Here, we assume that the mean climates of P1 and P2 are different to some extent because of global climate and environmental changes, which will be discussed further in Section 3. For the first type of reference period (Ref1), we calibrate the distribution of a specific PDF (Step 3 in Section 2.2) using the data from 1901 to 2014, which is used for estimating the SPE12 for the P1 and P2 estimation periods. For the second type of reference period (Ref2), calibrations are performed separately for P1 and P2, and thus so-called self-calibrated indices are derived. For the third type (Ref 3), we calibrate the distribution using the data from P1 (i.e., 1910–1957) and then use this distribution for both estimation periods.

**3 Results and discussion**

**3.1 Spatial and temporal patterns of climate variables**

Precipitation, air temperature and PET are investigated because they are used to estimate the SPEIs (Figs. 2 and 3 and Table 3). As noted already, the SPEIs are estimated based on the distribution of D (=P-PET in Eq. 1) and

here the air temperature is directly related to PET because we use the Thornthwaite approach to estimate PET. The selected regions show different climate features (Fig. 2), and EA and WA include the regions with a relatively wide range of mean precipitation from almost zero to more than 2000 mm per year. In terms of mean air temperature, it is clear that WA is generally quite warmer than other regions. Thus the relatively high PET in WA is clearly presented. Furthermore, the mean precipitation, air temperature and PET are quite similar between the CRU and UDEL.

To investigate the temporal changes of precipitation, air temperature and PET, we compared the means and the standard deviations between two periods (i.e., P1 and P2) in Table 3 and performed the Mann-Kendall trend test (Fig. 3). Table 3 presents clearly different temporal patterns for precipitation depending on the regions and all increasing temporal patterns for air temperature. Additionally, the annual precipitation in EA slightly decreased from 637.19 mm to 635.52 mm in the CRU (-0.2%) and from 659.67 mm to 649.21 mm in the UDEL (-1.6%). Moreover, in WA, the annual precipitation decreased substantially from 698.49 mm to 666.59 mm in the CRU (-4.6%) and from 734.84 mm to 676.11 mm in the UDEL (-8.0%). However, the annual precipitation increased in EU (25.17 mm (5.4%) in the CRU and 14.14 (3.5%) mm in the UDEL) and US (37.78 mm (3.7%) in the CRU and 24.92 mm (2.1%) in the UDEL).

For annual averaged air temperature, the averaged changes of air temperature between P1 and P2 in CRU (UDEL), which were 0.59 (0.37)°C in EA, 0.50 (0.27)°C in EU, 0.32 (0.05)°C in US and 0.35 (0.26)°C in WA, were generally greater than the differences between the CRU and UDEL. The annual averaged temperature became higher from the EA (6.16°C) to EU (6.99°C) to US (10.59°C) to WA (10.52°C) for P1. Consequently, the increasing ratios of annual averaged temperature in CRU (UDEL) were 9.70 (5.92)%, 7.18 (3.85)%, 3.06 (0.47)% and 1.33 (0.98)% in the EA, EU, US and WA, respectively. Such changes in air temperature directly influence the changes in PET, as we used the Thornthwaite approach for estimating PET.

For annual PET, the average growth amounts in the CRU and UDEL in P2 were higher than those in P2, in which increases are 17.09 mm and 9.08 mm in CRU and UDEL in EA, 26.42 mm and 14.20 mm in CRU and UDEL in EU, 17.11 mm and 2.42 mm in CRU and UDEL in US, 111.80 mm and 95.37 mm in CRU and UDEL in WA, because the air temperature is main factor to estimate the PET in this study. Consequently, the increasing ratios of annual PET in CRU (UDEL) were 2.5 (1.3)%, 4.4 (2.4)%, 2.4 (0.3)% and 5.9 (4.9)% in the EA, EU, US and WA, respectively.

The Mann-Kendall trend tests for annual precipitation, annual averaged temperature and annual PET were also performed, as shown in Fig. 3. The data reflect whether these variables showed statistically increasing, decreasing or no trends. For annual precipitation in EA, the areal extent with increasing trend was almost twice than that with a decreasing trend in the CRU, but the areal extent with a decreasing trend in the UDEL was broader than that with increasing area. In EU and US, the areal extent with an increasing trend was clearly greater than that with decreasing area in both the CRU and UDEL. However, in WA, the areal extent with a decreasing trend was larger than that with an increasing trend in both the CRU and UDEL. These patterns were usually more severe in the CRU than those in the UDEL. For annual averaged air temperature and PET, the CRU showed an increasing trend over most of the regions. Similar patterns were found in the UDEL, but the areal extent of the decreasing trend was slightly larger than that in the CRU.

**3.2 Temporal patterns of drought index**

The drought index (i.e., SPEI-12) is estimated for two periods of P1 and P2 with three different reference periods (Table 2) as described in Section 2.4. Fig. 5 shows the temporal variations of SPEI-12 depending on the reference periods (Ref1, Ref2 and Ref3) and datasets (CRU and UDEL) for the two periods. For US and EU, the SPEI-12 averages are very similar for the two periods: 0.005 (P1) and 0.118 (P2) in the US and -0.011 (P1) to -0.001 (P2) in EU. In EA, the SPEI-12 averages with the three different reference periods slightly decrease from P1 to P2, whereas the deviations of SPEI-12 increase markedly. In WA, the averages and deviations of SPEI-12 significantly decrease and increase, respectively, from P1 to P2. Furthermore, the variances of SPEI are relatively small for P1 compared to those for P2 in EA and WA while no noticeable differences in the variances are captured in EU and US. It may attribute to the lack of ground-based observations before 1950 (i.e., the most of P1) (Becker et al., 2013; Vittal et al., 2013; Nasrollahi et al., 2015) and such limit in data availability seems play a role in reducing the variance of SPEI for P1 in EA and WA. With regional averages, the role of the reference period is not clear and thus we investigate the spatial patterns of SPEI-12 hereafter.

Based on the Mann-Kendall trend test with annual SPEI-12 from 1901 to 2014, we present the increasing (i.e., wetting), decreasing (i.e., drying) or no trend over the regions (Fig. 6). First, the spatial distribution of SPEI-12 trends is identical between Ref1 and Ref3 and that in Ref2 is different. Ref1 and Ref2 use different calibration datasets but are similar in using one dataset for the two estimation periods; Ref2 uses a different calibration dataset for different estimation periods (Table 4). Therefore, SPEI-12 with Ref2 shows relatively less area with wetting and drying trends for the first and second periods relative to Ref1 and Ref2.

Regarding the temporal characteristics over different regions, the following are our findings based on Ref1 and Ref3: In WA, the drying trends are clearly dominant. In EU, the drying trends are scattered over the domain. In US, the wetting trends are scattered in the eastern region and the drying trends in the southwestern region. In EA, the drying trends are clearly in the western region.

Based on the grid-level trend analyses of precipitation, air temperature, PET and SPEI-12, we categorize each grid cell based on increasing, decreasing or neutral trends for each variable (i.e., precipitation, air temperature, PET and SPEI-12). For SPEI-12, increasing and decreasing trends represent wetting and drying trends. We present the ratio of each case relative to the total number of cases (i.e., total number of effective grid cells in all four regions), as shown in Fig. 7. First, the SPEI-12 trends are the same between Ref1 and Ref3, as the estimation periods share the one reference period in both Ref1 and Ref3 while each estimation period uses its own reference period in Ref2. Thus, the values of SPEI-12 are different in both cases, but the trends (i.e., relative values) are the same. Second, precipitation and air temperature exhibit neutral (or no) trends (in the center panel; presumably stationary climate), and the grid percentages of different trends in SPEI-12 vary between Ref1/Ref3 and Ref2. However, the ratio is relatively small, as most grid cells display increasing temperature and PET trends. Finally, based on neutral precipitation and increasing air temperature and PET trends in most grid cells, the numbers of cells with neutral and drying SPEI-12 trends are notably different between Ref1/Ref3 and Ref2. An increasing temperature and PET trend can be observed in most regions; thus, it is important to consider its impact on SPEI. It is particularly true as we use the Thornthwaite approach using the air temperature as a significant control variable of PET.

**3.3 Frequency, severity and spatial extent of drought**

In this section, we examine how the reference periods play a role in assessing the frequency, severity and spatial extent of drought using SPEI-12. The definitions of frequency, severity and spatial extent of drought used in this study are clarified in Section 2.3, and they may differ in different studies.

As explained above, a drought event is counted when the monthly SPEI-12 is estimated to be at or below -1.0 for the drought duration-frequency relation. For each drought event of grid cell, the duration is how long the SPEI-12 stays at or below -1. The frequency is the ratio between the total number of drought events and the effective grid points in each region (Fig. 8). We first find that the drought events with longer durations (prolonged right tails in the plot) occur more frequently in P2 than in P1 in all regions. However, we do not find any particular differences among the three different reference periods except in WA. The drought frequencies differ among the three reference periods. The frequencies with Ref2 and Ref3 are higher than those with Ref1 for P1, and slight differences in the frequency among the three reference periods are found around the 12-month duration for P2.

We examine how the severity of drought varies with the moving window sizes for the averaged monthly SPEI-12. Fig. 9 shows the severest SPEI-12 estimates, defined as the lowest value among the regional monthly average of SPEI-12 for the moving windows from 1 month to 12 months. In EU and US, we find no large differences among the SPEI-12s with Ref1, Ref2 and Ref3 for the same period. In these regions, the severest SPEI-12s for P1 are higher than those for P2. Such findings are seemingly inconsistent with the recently observed severe drought events in US and EU, but it is possible since we examine the regionally averaged indices, not the local extremes of SPEIs. Also it is consistent with Fig. 4. In US, the increase in precipitation is higher than that in PET, which leads the increase in D (=P-PET in Eq.1). In EU, the increase in PET is higher than that in precipitation, and thus the decrease in D is found in terms of average but the slight increase in the lower extreme of D is found. Therefore the severest drought events present less severe in P2 compared to those in P1. Nonetheless such changes in SPEI-12s according to the relative changes between P and PET suggests the important role of air temperature in drought severity in particular because the Thornwaite approach, using the monthly temperature as a major control variable for PET, is used to estimate the SPEI in this study.

In EA and WA, there exist different patterns in the severest SPEI-12s. The annual precipitation and air temperature (and thus PET) exhibit regionally scattered decreases and widespread increases, respectively (Fig. 3). Consequently, the droughts in 1958–2014 are more severe than those in P1. Furthermore, the severities vary significantly with the calibration periods in EA and WA, where the changes in precipitation and air temperature between two periods are marked.

The spatial extents of droughts for the annual SPEI-12 $\leq$ -1.0 are examined by sorting the results in ascending order (Fig. 10). We count the numbers of grid points with the SPEI-12 less than -1.0 for each period (i.e., P1 and P2) and divide them with the effective grid numbers in the region to derive the spatial extent, i.e., the grid percentage of droughts. Then the annual time series of spatial extent are sorted in ascending order, from the smallest to the largest. No specific patterns are evident for EU and US. In EA and WA, the spatial extents are generally broader in P2 than in P1. In particular, the spatial extents in 1958–2014 clearly diverge among the

different calibration periods, suggesting the importance of the calibration (i.e., reference periods in assessing the droughts in a region).

To understand how the drought characteristics would change if the reference period is dry or wet, we compare the drought spatial extent (%) for dry and wet cases in EA, EU, US and WA with defining dry and wet cases as below. We define the drought and wet cases with using a water surplus or deficit D (=P-PET in Eq. 1). We compare Ds between the reference period and estimation period. A value of D in the estimation period less than that in the reference period represents the dry case, i.e., the estimation period is drier than the reference period. We perform such analyses only in Ref1 for estimation periods of 1901-1957 (P1) and 1958-2014 (P2) and a reference/calibration period from 1901-2014 (P1+P2). For dry and wet cases, we quantify the spatial extent (%) according to the three different drought levels (D1, D2 and D3, which denote the cases of SPEI < -1.0, SPEI < -2.0 and SPEI < -3.0, respectively) in the four regions.

As presented in Table 5, the average D in P1 or P2 (estimation period) is smaller than that in P1+P2 (reference period), and it is considered to be the dry case. For example, in EA, the Ds values in P2 and P1+P2 are -4.89 mm/month and -5.07 mm/month, respectively; thus, it is a dry case. Then, for each case, the drought spatial extents, the number of drought grid cells relative to the total number of effective grid cell, are analyzed as shown in Fig. 11. The drought spatial extent tends to be larger in the dry case than that in the wet case in most regions, particularly in West Africa. However, we also note there are a few exceptions, which may be attributed to the fact that we use the regional average Ds. Thus, we cannot consider the grid-level variability in Ds.

**3.4 Case studies with historical drought events**

SPEI-12s with different reference periods are evaluated for historical drought events selected in each region to investigate how different reference periods influence the drought assessments of historical events. One drought event is chosen for each region as follows: 1) For East Asia, droughts that occurred in northern China in 2001 are chosen. These events caused economic losses of USD 1.52 billion (Zhang and Zhou, 2015). 2) For EU, we chose a 2003 drought that was caused by the European heat wave and spread over the majority of Europe (Stagge et al., 2013; Spinoni et al., 2015). 3) For US, we chose 2012 as the period of study as drought in that year was the most extensive drought over half of the US since the 1930s and caused economic losses of USD 31.2 billion (Smith and Katz, 2013; National Climate Data Center, 2015). 4) For West Africa, the drought in 1984 is chosen because it is one of severest droughts that has occurred over most Sahel countries (Gommes and Petrassi, 1994; Rojas et al., 2011; Masih et al., 2014).

By estimating SPEI-12 for a chosen year in each region, we compare the magnitudes of SPEIs (Figs. 12, 13, 14 and 15). Here the annual SPEI-12s with the monthly climate data from January to December in each year are first constructed and then the SPEI-12s for a chosen year are examined in detail. All SPEI-12s with the different reference periods present the drought status because we chose specific years with drought events. In general, all cases reveal that the SPEI-12 estimates in Ref2 are relatively high (i.e., wet) and those in Ref3 are relatively low (i.e., dry) for EA and WA, where drying temporal trends are clear. In particular, the several extremes (i.e., out of the scale ranges in Figs. 12-15) of SPEI-12 in Ref3 cases highlight the importance of the reference period. By using the reference period of the certain past time (P1 in this study, i.e., Ref3), the drought events in the

estimation period could be beyond the range in which the distribution is calibrated for the index. Essentially, using Ref3, it is assumed that not only the stationarity of the climate but also that the whole probability distribution of droughts is sampled in this period.

Furthermore, the percentages of drought spatial extent, i.e., the number of drought grid points relative to the total grid points, are assessed with different drought thresholds (Table 6). In most cases, the spatial extents of drought with the SPEI less than certain thresholds, such as -1, -2 or -3 (i.e., D1, D2 and D3 as in Table 1) are the greatest in Ref3 among the three cases with different reference periods. These results with the spatial extent are consistent with the results with the SPEI-12 estimates above. In addition, for the severe droughts with the drought events, defined with low thresholds such as SPEI-12 less than -2 or -3, greater percentages in Ref3 than in Ref1 and Ref2 are consistently obtained without exception in all regions of EA, EU, US and WA.

**4 Conclusions**

This study seeks to understand how a different reference period (i.e., calibration period) of climate data for estimating the drought index would influence the regional drought assessment. Specifically, we investigate the influence of different reference periods on historical drought characteristics such as trends, frequency, intensity and spatial extents using SPEI-12 from the CRU and UDEL datasets. For the 1901–1957 (P1) and 1958–2014 (P2) estimation periods, three different types of reference periods are used. For the first case, the data from 1901 to 2014 (P1+P2) are used for both estimation periods; for the second case, the data from P1 and P2 are used separately for the estimation periods of P1 and P2, respectively (self-calibrated); and for the final case, the data from P1 (1910–1957) are used for both estimation periods.

Focusing on the EA, EU, US and WA regions, we find the influence of the reference periods is significant in the regions with dominant drying trends from P1 to P2, such as EA and WA. It is also suggested that it is necessary to quantify the trends of climate variables such as precipitation and air temperature as the first step in selecting a reference period. We find that the reference periods influence the assessment of drought characteristics, particularly for severity and spatial extent, based on two datasets; however, their influence on the frequency is relatively small. Finally, self-calibration, the most common practice with an index such as SPEI, tends to underestimate the drought severity and spatial extent relative to the other approaches examined in this study.

This study highlights the need for the reference period to be clarified in drought assessments for a better understanding of regional drought characteristics and their temporal changes, particularly under climate change scenarios. Although this study with historical data may shows the different results depending on the selected local area, a similar study with historical data or climate change scenarios in different regions would undoubtedly strengthen our findings. We note that this study focuses on the temporal aspects of calibration data (i.e., calibration period). As briefly mentioned in the Introduction, using data from a particular station or grid, the averaged data for calibration would permit a meaningful comparison of the drought index among different locations. In conjunction with temporal considerations, such spatial issues could readily 
[revised manuscript text omitted]

[Figure]

**Figure 1. Study area and elevation investigated in this work.**

[Figure]

**Figure 2. Annual precipitation (mm), annual averaged temperature (°C) and annual PET (mm) for the CRU and UDEL datasets for P1 and their difference between P1 and P2.**

[Figure]

**Figure 3. Trends of annual precipitation, annual averaged temperature and annual PET for the CRU and UDEL datasets. PR and TA denote precipitation and temperature, respectively, and IN, N and DE indicate increasing, no trend and decreasing, respectively.**

(a) Annual precipitation          (b) Annual PET          (c) Annual surplus or deficit

[Figure]

**Figure 4. Temporal variations of annual precipitation, PET and deficit (D) depending on two datasets (CRU and UDEL) and periods (1901-1957 and 1958-2014). In the box plot, the center line represents the median value; the top and bottom of box represent the 25th and 75th percentile of the data, respectively; the dot represents the outlier.**

[Figure]

**Figure 5. Temporal variations of SPEI with 12-month lag for three different reference periods (Ref1, Ref2 and Ref3) for the CRU and UDEL datasets and the periods 1901–1957 and 1958–2014. In the box plot, the center line represents the median value; the top and bottom of box represent the 25$^{th}$ and 75$^{th}$ percentile of the data, respectively; the dot represents the outlier.**

[Figure]

**Figure 6. Trend of SPEI with 12-month lags (SPEI-12) for three different reference periods (Ref1, Ref2 and Ref3) for the (a) CRU and (b) UDEL datasets. WE, N and DR denote wetting, no trend and drying, respectively.**

[Figure]

(a-1) Precipitation vs. Air temperature (CRU)

(b-1) Precipitation vs. Air temperature (UDEL)

(a-2) Precipitation vs. PET (CRU)

(b-2) Precipitation vs. PET (UDEL)

**Figure 7. The SPEI trends with 12-month lag (SPEI12) for three different reference periods (Ref1 to Ref3) for the CRU and UDEL datasets based on the trends of monthly precipitation and temperature (or PET) in the four zones**

[Figure]

**Figure 8. Ratio of the number of drought events and effective data grid points for the CRU and UDEL datasets and the periods 1901–1957 and 1958–2014.**

[Figure]

**Figure 9.** Severest moving average of regional average SPEI for 1–12 months for three different reference periods (Ref1, Ref2 and Ref3) for the CRU and UDEL datasets and the periods 1901–1957 and 1958–2014.

[Figure]

**Figure 10. Spatial extent (%) for SPEI with 12-month lag < -1.0 for three different reference periods (Ref1, Ref2 and Ref3) for the CRU and UDEL datasets and the periods 1901–1957 and 1958–2014.**

(a) Monthly average D         (b) Averaged drought area (%)

[Figure]

[Figure]

**Figure 11. Monthly average D in Eq. (1) and averaged drought area depending on two datasets (CRU and UDEL) and four zones (EA, EU, US and WA) with the Ref1 condition (In Fig. 10 (a), all denotes the period for 1901-2014. In Fig. 10 (b), the dry status means that monthly average D in assessment period is less than those in reference period and the wet status denotes that monthly average D in assessment period is greater than those in reference period in the Ref1 condition.**

[Figure]

**Figure 12. SPEI with 12-month lag (SPEI12) for three different reference periods (Ref1, Ref2 and Ref3) for the (a) CRU and (b) UDEL datasets in East Asia in 2000.**

[Figure]

**Figure 13. SPEI with 12-month lag (SPEI12) for three different reference periods (Ref1, Ref2 and Ref3) for the (a) CRU and (b) UDEL datasets in Europe in 2003.**

[Figure]

**Figure 14. SPEI with 12-month lag (SPEI12) for three different reference periods (Ref1, Ref2 and Ref3) for the (a) CRU and (b) UDEL datasets in the United States in 2012.**

[Figure]

**Figure 15. SPEI with 12-month lag (SPEI12) for three different reference periods (Ref1, Ref2 and Ref3) for the (a) CRU and (b) UDEL datasets in West Africa in 1984.**

**Table 1. Classification of dry status in this study (Mckee et al., 1993).**

| Category | Description | SPEI |
|----------|-------------|------|
| D1 | Moderate dry | ≤ -1.0 |
| D2 | Extreme dry | ≤ -2.0 |
| D3 | Very extreme dry | ≤ -3.0 |

**Table 2. Climate variables and conditions for SPEI.**

| Type | Estimation Period | Calibration Period |
|------|-------------------|--------------------|
| **Ref1** | 1901−1957 | 1901−2014 |
|  | 1958−2014 |  |
| **Ref2** | 1901−1957 | 1901−1957 |
|  | 1958−2014 | 1958−2014 |
| **Ref3** | 1901−1957 | 1901−1957 |
|  | 1958−2014 |  |

**Table 3. Mean and standard deviation (STD) of precipitation and air temperature over the regions.**

| | | | CRU | | UDEL | |
|---|---|---|---|---|---|---|
| | | | 1901−1957 | 1958−2014 | 1901−1957 | 1958−2014 |
| Annual Precipitation (mm) | EA | Mean | 637.19 | 635.52 | 659.67 | 649.21 |
| | | STD | 22.36 | 30.05 | 30.67 | 31.76 |
| | EU | Mean | 685.86 | 711.03 | 674.17 | 688.31 |
| | | STD | 31.08 | 32.43 | 30.97 | 31.16 |
| | US | Mean | 698.44 | 736.22 | 709.50 | 734.42 |
| | | STD | 43.31 | 41.48 | 44.06 | 41.55 |
| | WA | Mean | 698.49 | 666.59 | 734.84 | 676.11 |
| | | STD | 36.87 | 43.84 | 44.89 | 48.00 |
| Air Temperature (°C) | EA | Mean | 6.08 | 6.67 | 6.25 | 6.62 |
| | | STD | 0.28 | 0.52 | 0.31 | 0.48 |
| | EU | Mean | 6.96 | 7.46 | 7.02 | 7.29 |
| | | STD | 0.56 | 0.68 | 0.55 | 0.64 |
| | US | Mean | 10.46 | 10.78 | 10.59 | 10.64 |
| | | STD | 0.45 | 0.50 | 0.43 | 0.43 |
| | WA | Mean | 26.27 | 26.62 | 26.40 | 26.66 |
| | | STD | 0.25 | 0.48 | 0.26 | 0.41 |
| Annual potential evapotranspiration (mm) | EA | Mean | 688.69 | 705.78 | 700.44 | 709.52 |
| | | STD | 16.06 | 23.92 | 15.65 | 22.73 |
| | EU | Mean | 598.06 | 624.48 | 603.15 | 617.35 |
| | | STD | 24.86 | 31.66 | 23.46 | 28.54 |
| | US | Mean | 711.49 | 728.60 | 718.48 | 720.90 |
| | | STD | 23.84 | 26.28 | 22.59 | 21.28 |
| | WA | Mean | 1889.57 | 2001.37 | 1948.91 | 2044.28 |
| | | STD | 72.61 | 136.96 | 78.17 | 129.94 |

**Table 4. Spatial extent (%), the number of grid points relative to the total effective grid point in each region for different trends with the different reference periods of SPEI-12.**

| Zone | CRU | | | | | | UDEL | | | | | |
| | Ref1 | | Ref2 | | Ref3 | | Ref1 | | Ref2 | | Ref3 | |
| | Wet | Dry | Wet | Dry | Wet | Dry | Wet | Dry | Wet | Dry | Wet | Dry |
| --- | --- | --- | --- | --- | --- | --- | --- | --- | --- | --- | --- | --- |
| EA | 2.5 | 36.3 | 0.0 | 8.0 | 2.5 | 36.5 | 3.4 | 23.2 | 0.0 | 7.7 | 3.4 | 23.2 |
| EU | 10.4 | 24.9 | 0.0 | 1.7 | 10.4 | 24.9 | 5.3 | 15.8 | 0.0 | 2.2 | 5.3 | 15.8 |
| US | 18.6 | 16.2 | 0.0 | 67 | 18.6 | 16.2 | 11.3 | 9.7 | 0.1 | 3.1 | 11.3 | 9.7 |
| WA | 0.0 | 90.2 | 0.0 | 40.4 | 0.1 | 89.8 | 0.0 | 90.9 | 0.0 | 19.5 | 0.0 | 90.9 |

**Table 5. Monthly average D (mm/month) in four study regions.**

| | CRU | | | UDEL | | |
|---|---|---|---|---|---|---|
| | P1 | P2 | P1+P2 | P1 | P2 | P1+P2 |
| EA | -4.29 | -5.85 | -5.07 | -3.40 | -5.03 | -4.21 |
| EU | 7.32 | 7.21 | 7.26 | 5.92 | 5.91 | 5.92 |
| US | -1.09 | 0.64 | -0.23 | -0.75 | 1.13 | 0.19 |
| WA | -99.26 | -111.23 | -105.24 | -101.17 | -114.01 | -107.59 |

**Table 6. Spatial extent (%) (the number of grid points belonging to each drought category, relative to the total grid point) for the major drought events.**

| Zone | Period | Type | CRU | | | UDEL | | |
|------|--------|------|------|------|------|------|------|------|
| | | | Ref1 | Ref2 | Ref3 | Ref1 | Ref2 | Ref3 |
| EA | 2000 | D1 | 32.63 | 27.48 | 38.80 | 26.81 | 27.39 | 29.62 |
| | | D2 | 2.45 | 0.75 | 14.64 | 0.92 | 0.73 | 2.64 |
| | | D3 | 0.05 | 0.00 | 1.83 | 0.04 | 0.01 | 0.07 |
| EU | 2003 | D1 | 37.58 | 39.10 | 36.68 | 35.30 | 34.67 | 36.61 |
| | | D2 | 5.33 | 3.97 | 7.68 | 5.93 | 4.82 | 8.50 |
| | | D3 | 0.00 | 0.00 | 0.02 | 0.02 | 0.10 | 0.22 |
| US | 2012 | D1 | 52.16 | 55.01 | 50.02 | 54.69 | 56.32 | 52.92 |
| | | D2 | 11.97 | 11.90 | 15.74 | 10.36 | 11.76 | 11.63 |
| | | D3 | 0.02 | 0.00 | 0.53 | 0.09 | 0.05 | 0.87 |
| WA | 1984 | D1 | 44.06 | 31.04 | 62.18 | 37.13 | 27.15 | 57.78 |
| | | D2 | 3.42 | 1.87 | 28.62 | 2.07 | 1.72 | 13.80 |
| | | D3 | 0.00 | 0.00 | 14.30 | 0.00 | 0.00 | 2.99 |

---

## Referee Report (RR1)

Specific comments:

1.  According to Dai and Zhao (2016, Climatic Change, doi:10.1007/s10584-016-1705-2.), the CRU precipitation data ha a series quality issue over many land areas with higher elevation and or mountains, in particular, over arid regions, where the gaps is always filled with data from far away stations or with 1961-1990 climatology, especially for the recent decades. That is why there is no trend in CRU for manly land areas as shown in Fig. 3a. Although the UDEL is also used to compare with the CRU for climatological mean and long-term trends, the comparison of the long-term temporal variation is not conducted, which is important to address the temporal difference between two datasets since about 1950's.

2.  Dai and Zhao (2016), Zhao et al (2014) and other studies, the GPCC V6 or V7 product is better than CRU precipitation data to describe spatio-temporal variations and variability over global land areas, especially over the arid –semiarid regions. So, I suggest the authors use the GPCC, CRU, and the UDEL precipitation products but with same temperature data to perform same analysis, and then compare the impacts of the different precipitation products on the different calibration periods of SPEI.

3.  Some other recent studies discussed the drought change and variations are still needed to cited, such as Zhao and Dai (2015, J. Climate, doi:10.1175/JCLI-D-14-00363.1.; 2016, Climatic Change, doi:10.1007/s10584-016-1742-x.).

4.  The title should be clarified the drought index of the SPEI because this manuscript only focuses on the SPEI but not on other drought index.

5.  Line 21, P1, 'limited to' should be 'limited by'.

6.  Line 20-21, P1, "Although" has been used in this sentence, why is "nevertheless" still used?

7.  Line 31, P1, 'stated', is not 'state'.

8.  Line 31, P2, 'This study' would be revised as 'our study' to clarify the following description is the main findings obtained from this manuscript.

9.  'Data and method' is better for the title of the section 2.

10. There are large gaps for the English writing for this manuscript.

---

## Author Response (AR2)

Response letter to reviewer's comments

Please find our responses to the reviewer's comments in blue font below along with the track-changed revised manuscript.

**Review by Editor**

I struggled with a decision on this manuscript. Reviewer #1 has recommended that the manuscript be published in its present form; whereas, the Reviewer #2 rated the manuscript as 'poor' in all categories and recommended continued revision with further review. Reviewer #2 also did not feel their comments were fully responded to and some responses led to additional questions. Despite this, Reviewer #2 does note however, that the manuscript would provide a welcome contribution to the literature if their comments were fully addressed and parts of the manuscript were clarified.

I have carefully read the revised manuscript and agree with the comments of Reviewer #2. Those comments should be addressed as well as the additional comments below. The manuscript will need to be further revised and will be sent out for additional review comments. Furthermore, the manuscript can still be rejected if there are questions that remain in the next round of revisions.

At this point in the process, I direct the authors to address the comments of Reviewer #2 and also my comments below:

- Are L-moments commonly used to fit the SPEI? If not, please also cite other methods of parameter estimation that are used in the literature. The manuscript also provides no information of the goodness of fit or estimated parameters of the 3-parameter log-logistic for the readers to assess if this were a reasonable assumption or replicate the analysis.

>> The L-moments approach has become commonly used to fit the SPEI since its first proposal and application by Vicente-Serrano et al. (2010). In the revised manuscript, we have added L-moment ratio diagrams to provide information on the goodness of fit of the three-parameter log-logistic model. We have presented the results with CRU data for the reference period of 1901-2014 as an example.

Page 6: *"The drought index (i.e., SPEI-12) was estimated by fitting the three-parameter log-logistic model for three different reference periods (Table 2), as described in Section 2.4. As shown in the L-moment ratio diagram with CRU for Ref1 as an example (Fig. 5), the model is well fitted with the L-moment approach, following Vicente-Serrano et al. (2010)."*

[Figure]

**Figure 5. L-moment ratio diagrams for D in Eq. (1) with a 12-month timescale based on CRU for each region for 1901-2014, 1901-1957.**

- p. 4, It is not necessary to explain the details of the Mann-Kendall test if the reference is provided. This test is common enough that detailed explanation is not needed. What was the significance level selected for this study? This needs to be noted in the text and Figure 3.

>> We have removed the details of the Mann-Kendall test and specified the significance level of the test as follows.

In 2.3 Temporal trends and statistical characteristics:
*"The temporal trend is investigated with a nonparametric and monotonic trend test based on the S-statistic of the Mann-Kendall trend test (Mann, 1945; and Kendall, 1976). In this test, an increasing (positive) trend or decreasing (negative) trend is tested for at a significance level of 5%."*

Figure 4:
*"Trends in annual precipitation, annual averaged temperature and annual PET based on the CRU and UDEL datasets. PR and TA denote precipitation and temperature, respectively, and IN, N and DE indicate statistically increasing (positive) trend, no trend and decreasing (negative) trend, respectively, with the significance level of 5%."*

- Section 3.1: The link between this section and the goals of the study are not clearly linked. I understand that they are used to estimate SPEI but there needs to be a more explicit link between this section and the hypotheses tested.
- Also in Section 3.1, many of the differences reported seem quite small (p. 6). The authors need to add more interpretation rather than just spending nearly a page listing the amounts by which the climate variables differ.
- At the end of Section 3.1, there needs to be figure and/or table references to support the statements being made. This is a larger issue with the manuscript that Reviewer #2 also points out - statements need to be backed up by a reference to results or by a citation.

>> Per the editor's suggestion, we have carefully reviewed Section 3.1 and revised the paragraphs (1) to clarify the purpose of this section regarding the changes of P, T and PET that are used to estimate SPEI and (2) to interpret the differences of climate variables while avoiding simple lists of numbers. In addition, the figures and tables (i.e., Figs 2, 3 and 4 and Table 3) in this section have further been referred in other sections to support our statements about the SPEI in Sections 3.2 and 3.3.

Page 5: *"In this section, we examine the spatial and temporal variations of precipitation, air temperature and PET (Figs. 2, 3 and 4 and Table 3), which are used to estimate D (= P - PET) (in Eq. 1) and thus the SPEI values. We particularly focus on the differences in meteorological conditions between P1 and P2 to enhance our understanding of similar or different drought index values according to the different reference periods in the following sections.*
*To investigate the temporal changes in precipitation, air temperature and PET, we compared the means and standard deviations between the two periods (i.e., P1 and P2) (Figs. 2 and 3 and Table 3). Most cases showed largely consistent results between CRU and UDEL; therefore, we did not focus extensively on the differences between the two datasets. In general, the temporal pattern of precipitation varied among regions, and increased air temperature was observed in all regions. On average (Fig. 3 and Table 3), annual precipitation was decreased in P2 relative to P1, as in EA and WA, whereas decreased precipitation was clearly evident only in limited areas within the regions (Fig. 2); for example, the west Sahel within WA. In contrast, annual precipitation increased in EU and the US. Increases in air temperature were clearly shown in all regions; consequently, increases in PET, which is controlled mainly by air temperature, were generally evident. Decreases in D were observed only in EA and WA (Fig. 4c). In these regions, an annual water deficit (i.e., negative D) was evident, whereas in other regions, i.e., EU and the US, an annual water surplus (i.e., positive D) was present."*

- You repeat the same statement about the Thornwaite approach three times (p. 5, line 35; p. 6, line 20; p. 8, line 38) but offer no reasons for choosing this approach. Also, how might the use of the another estimation method affected the results and conclusions?

>> We have removed repeated statements regarding the Thornthwaite approach and have added a sentence to address how a different estimation method might have influenced the results and conclusions.

Page 10: *"Furthermore, we noted that the Thornthwaite approach, in which air temperature is a main controlling factor of PET, is used to estimate SPEI in this study; however, other approaches such as the Penman method could be used to consider changes in other meteorological variables, such as wind, atmospheric humidity and radiation. McVicar et al. (2012) have suggested that there may be limited effects of temperature increase on drought through increased PET because other meteorological conditions affecting PET may compensate for the temperature increase."*

- Section 3.2: Again here evidence is needed, particularly in paragraph 1 and last paragraph on p. 7, to support these statements.

>> We have revised the first and last paragraphs of Section 3.2 to include additional results (Fig. 5) and references to the supporting figures.

Page 6: "*The drought index (i.e., SPEI-12) was estimated by fitting the three-parameter log-logistic model for three different reference periods (Table 2), as described in Section 2.4. As shown in the L-moment ratio diagram with the CRU for Ref1 as an example (Fig. 5), the model is well fitted with the L-moment approach, following Vicente-Serrano et al. (2010). ... As suggested in previous studies (i.e., Becker et al., 2013; Vittal et al., 2013; Nasrollahi et al., 2015), the limited availability of data for the early 20$^{th}$ century can result in underestimates of the spatial variabilities of climate variables in global datasets; in the present study, such limited data availability might have contributed to the reduced SPEI variance in P1 in EA and WA. Based on regional averages, the role of the reference period is not clear; thus, we investigated the spatial patterns of SPEI-12 hereafter.*"

Page 7: *"Second, precipitation and air temperature exhibit neutral (or no) trends (i.e., the center panel among the 3 x 3 panels in Fig. 8, indicating a presumably stationary climate), and the grid percentages of different trends in SPEI-12 vary between Ref1/Ref3 and Ref2. ... Finally, in the case of neutral precipitation and increasing air temperature (or PET) trends (i.e., the top middle panel among the 3 x 3 panels in Fig. 8), the numbers of cells with neutral and drying SPEI-12 trends are notably different between Ref1/Ref3 and Ref2. We observed increasing temperature and thus increasing PET trends in most regions (refer to Fig. 4). This discrepancy between the reference periods might play a critical role in assessing drought status."*

- While the study examines 3 reference periods, this is by no means exhaustive. Therefore, I agree with Reviewer #2 that the authors need to be careful to not overstate the conclusions. A much more rigorous testing would examine the effects of sample size and test many different combinations of reference periods to make broader

conclusions. As Reviewer #2 stated, the work could still be a useful contribution as it stands - but the conclusions need to be tempered to reflect that only 3 different reference periods were tested and sample size was not evaluated.

>> We agree with this comment. Accordingly, we have added a statement regarding the limitations of our study.

Page 10: *"However, we noted these results are drawn from only three sets of reference periods, two different datasets (i.e., CRU and UDEL) and four regional examples. Therefore, future work should evaluate different combinations of reference periods with increased sample sizes and with different datasets."*

- I agree with Reviewer #2 that Section 3.4 offers some nice observations.

>> Thank you for the comment.

- Tables and figures must be stand-alone. They currently do not meet the standards of HESS. Please see http://www.hydrology-and-earth-system-sciences.net/for_authors/manuscript_preparation.html to ensure compliance.

>> We have revised the figure and table titles accordingly; some examples are provided below.

*"Figure 1. Study area, including East Asia (EA), Europe (EU), the United States (US) and West Africa (WA), and elevation (m above sea level (a.s.l.)). The dashed blue box represents the boundary for each study region."*

*"Figure 2. Annual precipitation (mm), annual average temperature (°C) and annual potential evapotranspiration (PET) (mm) based on the datasets of the Climate Research Unit (CRU) and the University of Delaware (UDEL) for the period 1901-1957 (P1) and the difference between P1 and 1958-2014 (P2) (i.e., P2-P1)."*

*"Figure 8. The ratios of SPEI-12 trends for three different reference periods (Ref1 to Ref3 in Table 2) based on the CRU and UDEL datasets for trends of monthly precipitation and temperature (or PET) in each region. For SPEI, WET, N and DRY indicate statistically positive (wetting) trend, no trend and negative (drying) trend, respectively, at a significance level of 5%. For precipitation, air temperature and PET, IN, N and DE denote statistically increasing (positive) trend, no trend and decreasing (negative) trend, respectively, at a significance level of 5%."*

Minor comments:
p. 2, line 3: Do the authors mean only the drought indices noted in the Vicente-Serrano references or all drought indices?

>> We are referring to any drought indices with relative measures. We have revised the sentence for clarity.

Page 2: *"Relative drought indices, however, are limited in their utility because they are based on standardized or normalized shortages relative to average conditions at a*

*given station or in a specific period (Vicente-Serrano and Beguería-Portugués, 2003; Vicente-Serrano et al., 2010)."*

**Anonymous Referee # 2**

The study looks into the effects of the choice of reference period in standardizing the SPEI drought index.

By quantifying drought severity, duration and extent in maps and by aggregating SPEI values over four regions, conclusions are reached on 'best practice' in setting the calibration period. A very nice touch is that the authors have looked at specific record-dry years for the regions under consideration, and visualised the effects of the choice of calibration period on the drought estimate.

The SPEI, like many other drought metrics, is a standardised metric making its estimates for dry or wet conditions comparable over diverse climatological areas. The issue what calibration period to use, and the effects of not using the full-length of the period for which data is available as calibration, has been debated in the literature. I still think that this makes the study a very welcome contribution to the discussion. However, my view on the manuscript is still not changed. It raises more questions than answers.

Some of my main concerns have not been addressed sufficiently and some of the answers are difficult to understand.

My advise to the editor is to reconsider after major revisions.

Regarding my main concerns:

1) I now understand what the authors mean with "drought frequency", and especially their "effective grid points". The effective grid points are the grid boxes that have seen at least one drought event - irrespective of the length of the drought. Perhaps the authors could use a formulation like the one provided in this review.

>> As suggested by the reviewer, we have clarified our definitions of the drought measures.

Page 4: *"In this study, we defined three measures of drought based on the SPEI-12: (1) Drought frequency was calculated as the ratio of the total number of drought events (i.e., SPEI-12 ≤ -1) to the total number of terrestrial grid points; here, we counted the number of drought events without considering whether a given drought event (i.e., SPEI-12 ≤ -1) was identified consecutively. (2) Severity was defined as the lowest estimate of the regional monthly average SPEI-12 with moving windows with periods of 1 to 12 months; here, regional averages were estimated in the four study regions depicted in Fig. 1. (3) Spatial extent was calculated as the number of grid points with an annual SPEI-12 ≤ -1.0 relative to the total number of terrestrial grid points."*

2) fig 9 (7 in the original ms.) is still not understandable. I am still not convinced by the analysis. Lets take Europe and the CRU data. What you show in the figure is that (averaged over Europe) and for the 1901-1957 period, the lowest 1-month SPEI12 is lower than the lowest 1-month SPEI12 for the 1958-2014. As explanation, you give: "the increase in PET is higher than that in precipitation; thus, D decreases in terms of the average, but the lower extreme of D slightly increases. Therefore, the most severe drought events are less severe in P2 compared to those in P1." When Precipitation increase less fast than PET, then a trend towards drier conditions is expected - which is totally opposite to what you show. Furthermore, you claim that 'the lower extreme of D slightly increases' but this the observation you need to explain! Can you show which month in the 1901-1957 period has this low SPEI12 value? How does the spatial map of Europe look like during is exactly this drought? The only possibility I see is that the drought of the early 20th century relates to this low SPEI12 value. With the sparse network which is present in this period, the drought could be very extensive in the CRU TS data.

>> To investigate the possibilities raised by the reviewer, we have presented the events showing the severest SPEI in Table R1 and Figure R1 for EU with CRU as an example. As proposed by the reviewer, Figure R1 shows that the severest case in P1 (i.e., 1901-1957) appears to be more widespread than that in P2 (i.e., 1958-2014). Therefore, we have revised the text to include this information and to note that our results might be unrealistic due to the limitations of the datasets used in this study.

Page 8: *"However, at the lower extreme of D in this case (i.e., the lower extent of the vertical line in the box plot of D in Fig. 3c), a slight increase is apparent, indicating that the most severe drought events are less severe in P2 than in P1. In examining the spatial maps of the severest cases (not shown), we find that the severest drought event in P1 is more widespread than that in P2 in this case. Such widespread drought might be due to the sparse network of meteorological stations during the early 20th century, a possibility that awaits further study."*

Page 10: *"However, we noted these results are drawn from only three sets of reference periods, two different datasets (i.e., CRU and UDEL) and four regional examples. Therefore, future work should evaluate different combinations of reference periods with increased sample sizes and different datasets."*

**Table R1. Drought event of severest regional averaged SPEI-12 with the 1 month moving window for 1901-1957 and 1958-2014 in EU with CRU.**

| Reference | Event | |
|---|---|---|
| | 1901-1957 | 1958-2014 |
| Ref1 | 1946.11-1947.10 | 2010.12-2011.11 |
| Ref2 | 1946.11-1947.10 | 2010.12-2011.11 |
| Ref3 | 1946.11-1947.10 | 2002.12-2003.11 |

[Figure]

**Figure R1. Spatial distribution of SPEI-12 for the case listed in Table R1 (i.e., severest regional averaged SPEI-12 with the 1 month moving window for (a) 1901-1957 and (b) 1958-2014 in EU with CRU).**

The increase in value of SPEI12 with longer running mean is easy to explain. In the reply to the reviewer you give as explanation: "We therefore argue that the continuous upward trend from P1 to P2 is observed for the following reasons. First, we examine the regionally averaged indices, which do not necessarily capture local severe drought events. Second, we find that the trend is consistent with the results shown in Fig. 4."
I don't really understand what you write here, but the explanation is simple. You calculate SPEI12 averaged over e.g. Europe. First, you look to the lowest value SPEI12 attains - this is the 1-month value in your graphs. Then, you apply a 2-month running mean. This 2-month mean MUST be less negative than the 1-month mean, and so on with the 3 month mean etc. This is the reason why you find increases in all graphs with increasing window length. The slight downturn in east Asia is curious....

>> To clarify how we estimated the severest SPEI with the increasing window size, please refer to Figure R2 below. A lower SPEI-12 with an increasing moving period is possible because it is not necessary to include the severest SPEI-12 for one month in estimating the severest SPEI-12 for two months. For this reason, a slight downturn of the severest SPEI-12 with increasing window size was observed in some cases.

[Figure]

**Figure R2. Conceptual explanation how to estimate the severest SPEI-12 for different window sizes.**

Apart from these newly inserted 'explanations' I also find conclusions in the Conclusion section like: "Finally, self-calibration, the most common practice associated with indexes such as the SPEI, tends to underestimate the drought severity and spatial extent relative to the other approaches examined in this study.". I haven't found any evidence for this in paper....

>> According to this comment, we have revised the text to clarify our findings and limitations.

Page 10: *"Finally, we found that the use of the calibrated distribution with the past observations (i.e., Ref3) tends to overestimate the drought severity and spatial extent relative to the other approaches used in this study. However, we note these results are drawn from only three sets of reference periods, two different datasets (i.e., CRU and*

*UDEL) and four regional examples. Therefore, future work should evaluate different combinations of reference periods with increased sample sizes and with different datasets."*

Smaller issues:

page 5, line 8: replace "status" with "threshold"

>> We have revised the text accordingly.

page 5, lines 9-11 (Additionally, the spatial extent of drought was defined as the percentage of grid points below the given drought level, in which the corresponding drought index was less than the given drought category in each month.) Is the percentage calculated as the number of grid boxes below a threshold, with respect to the total number of gridboxes in the domain?

>> Accordingly, we have clarified the definition of the spatial extent of drought.

Page 4: *"Additionally, the spatial extent of drought was defined as the percentage of grid points below the given drought level, in which the corresponding drought index was less than the given drought category in each month, relative to the total number of terrestrial grid points in the domain."*

Table 1: please correct the description according to the references given in my first review (-2 < SPEI <= -1: moderately dry, -3< SPEI <= -2: severely dry, SPEI<=-3: extremely dry)

>> We have re-defined drought status based on the reference but with modification. For example, D1 includes all cases with an SPEI value below -1, which is different from the range from -2 and -1 as used in the referenced work. This has been clarified in the revised manuscript.

[revised manuscript text omitted]

Moved (insertion) [2]

(b-1) Trend of precipitation (UDEL)
(a-2) Trend of air temperature (CRU)
(b-2) Trend of air temperature (UDEL)
(a-3) Trend of PET (CRU)
(b-3) Trend of PET (UDEL)

[Figure]

Figure 4. Trends in annual precipitation, annual averaged temperature and annual PET based on the CRU and UDEL datasets. The colored regions correspond to regions of IN, N and DE, which indicate statistically positive (increasing) trend, no trend and negative (decreasing) trend, respectively, with a significance level of 5%.

[Figure]

L-moment ratio diagrams for D in Eq. (1) with a 12-month timescale based on CRU for each region for 1901-2014 and 1901-1957.

[Figure]

**Figure 6.** Temporal variations in SPEI-12 for three different reference periods (Ref1, Ref2 and Ref3 in Table 2) based on the CRU and UDEL datasets from 1901–1957 and 1958–2014. In the box plots, the center line represents the median value; the top and bottom of each box represent the 25th and 75th percentile of the data, respectively; and the dots represent outliers.

[Figure]

[Figure]

**Figure 7. SPEI-12 trends in three different reference periods (Ref1, Ref2 and Ref3 in Table 2) based on the (a) CRU and (b) UDEL datasets. The colored regions correspond to regions of WE, N and DR, which denote statistically positive (wetting) trend, no trend and negative (drying) trend, respectively, at a significance level of 5%.**

[Figure]

(a-1) Precipitation vs. Air temperature (CRU)    (b-1) Precipitation vs. Air temperature (UDEL)

(a-2) Precipitation vs. PET (CRU)    (b-2) Precipitation vs. PET (UDEL)

**Figure 8. The ratios of SPEI-12 trends for three different reference periods (Ref1 to Ref3 in Table 2) based on the CRU and UDEL datasets for trends of monthly precipitation and temperature (or PET) in each region. For SPEI, WET, N and DRY indicate statistically positive (wetting) trend, no trend and negative (drying) trend, respectively, at a significance level of 5%. For precipitation, air temperature and PET, IN, N and DE denote statistically increasing (positive) trend, no trend and decreasing (negative) trend, respectively, at a significance level of 5%.**

Formatted Table

[Figure]

**Figure 9.** Ratio between the number of drought events and the number of terrestrial data grid points for three different reference periods (Ref1, Ref2 and Ref3 in Table 2) based on the CRU and UDEL datasets from 1901–1957 and 1958–2014 in each region.

Moved (insertion) [4]

[Figure]

**Figure 10.** Most severe moving average of regional SPEI-12 for 1–12 months for three different reference periods (Ref1, Ref2 and Ref3 in Table 2) based on the CRU and UDEL datasets from 1901–1957 and 1958–2014 in each region.

[Figure]

**Figure 11. Spatial extent (%) of SPEI-12 < -1.0 for three different reference periods (Ref1, Ref2 and Ref3 in Table 2) based on the CRU and UDEL datasets from 1901–1957 and 1958–2014 in each region.**

[Figure]

[Figure]

(a) Monthly average D     (b) Averaged drought area (%)

**Figure 12. Monthly average D (mm) in Eq. (1) and average drought area (%) based on the CRU and UDEL datasets for each region of EA, EU, US and WA for the Ref1 condition. In (a), ALL denotes the period of 1901-2014. In (b), Dry denotes** that the monthly average D in the assessment period is less than that in the reference period, and **Wet denotes** that the monthly average D in the assessment period is greater than that in the reference period based on the Ref1 condition.

Formatted Table

Moved up [1]: Fig.

[Figure]

(a-1) SPEI-12 for 2000 with Ref1 (CRU)    (b-1) SPEI-12 for 2000 with Ref1 (UDEL)

(a-2) SPEI-12 for 2000 with Ref2 (CRU)    (b-2) SPEI-12 for 2000 with Ref2 (UDEL)

(a-3) SPEI-12 for 2000 with Ref3 (CRU)    (b-3) SPEI-12 for 2000 with Ref3 (UDEL)

**Figure 13. Spatial distribution of SPEI-12 for three different reference periods (Ref1, Ref2 and Ref3 in Table 2) based on the (a) CRU and (b) UDEL datasets in East Asia in 2000 as an example.**

[Figure]

... [12]

(a-1) SPEI-12 for 2003 with Ref1 (CRU)    (b-1) SPEI-12 for 2003 with Ref1 (UDEL)

[Figure]

[Figure]

(a-2) SPEI-12 for 2003 with Ref2 (CRU)    (b-2) SPEI-12 for 2003 with Ref2 (UDEL)

(a-3) SPEI-12 for 2003 with Ref3 (CRU)    (b-3) SPEI-12 for 2003 with Ref3 (UDEL)

**Figure 14. Spatial distribution of SPEI-12 for three different reference periods (Ref1, Ref2 and Ref3 in Table 2) based on the (a) CRU and (b) UDEL datasets in Europe in 2003 as an example.**

... [13]

(a-1) SPEI-12 for 2012 with Ref1 (CRU)    (b-1) SPEI-12 for 2012 with Ref1 (UDEL)

[Figure]

(a-2) SPEI-12 for 2012 with Ref2 (CRU)    (b-2) SPEI-12 for 2012 with Ref2 (UDEL)

(a-3) SPEI-12 for 2012 with Ref3 (CRU)    (b-3) SPEI-12 for 2012 with Ref3 (UDEL)

[revised manuscript text omitted]

**Formatted Table**

| Zone | Period | Type | CRU | | | UDEL | | |
|------|--------|------|------|------|------|------|------|------|
| | | | Ref1 | Ref2 | Ref3 | Ref1 | Ref2 | Ref3 |

---

## Author Response (AR3)

Please find the responses to the Editor's and reviewers' comments given in blue font, and the track-changed revised manuscript.

**Review by the Editor (Editor decision: publish subject to revisions)**

**Comments to the Author:**
I would like to very much thank the reviewers for their comments on this revised manuscript and the authors for their persistence in accounting for the review comments.

The manuscript has now gone through a second round of revisions. Referee #3 notes that there is still a gap needed to improve the paper for publication; Referee #4 notes that the manuscript should be accepted subject to minor revisions. Both reviewers note - as did the previous reviewers - that the subject of the manuscript and its findings have the potential to make a useful and interesting contribution. It is for this reason that I believe the manuscript should continue in the review process.

I would ask that the authors now respond to the referee comments focusing particularly on the two gaps that are noted - one each by Referee #3 and #4. I will send the manuscript out to for final review to ensure that all comments have been addressed.

I look forward to reviewing the revision.

Kindly,
Stacey

>> As suggested by the Editor and reviewers, we have revised the manuscript and focused on the uncertainty of the climate datasets used in the study (Referee #3), as well as the implications of our findings (Referee #4). Please refer to the responses to the reviewers' comments below.

**Anonymous Reviewer # 3 (Report #1)**

**Specific comments:**
1. According to Dai and Zhao (2016, Climatic Change, doi:10.1007/s10584-016-1705-2.), the CRU precipitation data ha a series quality issue over many land areas with higher elevation and or mountains, in particular, over arid regions, where the gaps is always filled with data from far away stations or with 1961-1990 climatology, especially for the recent decades. That is why there is no trend in CRU for manly land areas as shown in Fig. 3a. Although the UDEL is also used to compare with the CRU for climatological mean and long-term trends, the comparison of the long-term temporal variation is not conducted, which is important to address the temporal difference between two datasets since about 1950's.

Dai A, Zhao T (2016) Uncertainties in historical changes and future projections of drought. Part I: estimates of historical drought changes. Climatic Change. DOI: 10.1007/s10584-016-1705-2

>> As Dai and Zhao (2016) noted, the limits of CRU TS3.10.01, which include poor gauge coverage, are discussed. Additionally, the issues with CRU TS3.10.01 that were improved in CRU TS3.22 were noted. Moreover, we examined the CRU and UDEL precipitation datasets used in this study, as shown in Fig. R1. Indeed, we did not detect any significant problems in the CRU precipitation data in Fig. R1 because this study used CRU TS3.23, which is the updated version of CRU TS3.22 that is used in Dai and Zhao (2016). Note that we have corrected a mistake in the original manuscript and changed CRU TS3.10 to CRU TS3.23. In summary, we did not find any significant or erroneous temporal differences between the CRU and UDEL datasets. In the revised manuscript, we have added the findings of Dai and Zhao (2016) and justified our usage of the CRU and UDEL datasets as follows.

Page 2: *"Dai and Zhao (2016) examined uncertainties in the sc-PDSI due to different choices of forcing data and the calibration period. They recommend using the Global Precipitation Climatology Center (GPCC) or the Global Precipitation Climatology (GPCP) datasets over other existing land precipitation products, such as CRU, and not including years after 1980 in the calibration period due to the influence of anthropogenic climate change."*

Page 3: *"As briefly noted in Section 1, Dai and Zhao (2016) suggested that the GPCC or GPCP dataset should be used instead of the CRU datasets in drought assessment with the sc-PDSI. They noted the limitations of the CRU dataset (specifically CRU TS3.10.01) due to its poor data coverage since the 1990s. In this study, the CRU TS3.23 and the UDEL datasets are used because these datasets provide both precipitation and temperature data, whereas the GPCC and GPCP datasets include only precipitation data."*

[Figure]

Figure R1. Temporal changes in regionally averaged precipitation based on the CRU and UDEL datasets. The dashed lines indicate the 95% confidence interval.

2. Dai and Zhao (2016), Zhao et al (2014) and other studies, the GPCC V6 or V7 product is better than CRU precipitation data to describe spatio-temporal variations and variability over global land areas, especially over the arid –semiarid regions. So, I suggest the authors use the GPCC, CRU, and the UDEL precipitation products but with same temperature data to perform same analysis, and then compare the impacts of the different precipitation products on the different calibration periods of SPEI.

Zhao T, Chen L, Zhuguo M (2014) Simulation of historical and projected climate change in arid and semiarid areas by CMIP5 models. Chinese Science Bulletin 59(4): 412-429.

>> We agree that the suggested approach by the reviewer could be used to investigate the effects of the different precipitation products on the different calibration periods of SPEI. However, it is beyond the focus of this study; thus, we have added this investigation as a possible point of future study (see Section 4).

Page 10: *"However, we note that the abovementioned results are drawn from only three sets of reference periods, two different datasets (i.e., CRU and UDEL) and four regional examples. Future work should evaluate different combinations of reference periods with increased sample sizes and different datasets. The combined datasets could also be used to focus on the effects of different precipitation or temperature products on the SPEI. For example, the precipitation data from the CRU, UDEL and GPCC, which is suggested to be better than CRU data by Dai and Zhao (2016), and the temperature data from CRU could be utilized to focus on the effects of different precipitation products."*

3. Some other recent studies discussed the drought change and variations are still needed to cited, such as Zhao and Dai (2015, J. Climate, doi:10.1175/JCLI-D-14-00363.1.; 2016, Climatic Change, doi:10.1007/s10584-016-1742-x.).

Zhao T, Dai A (2015) The magnitude and causes of global drought changes in the twenty-first century under a low–low-moderate emissions scenario. Journal of Climate 28: 4490–4512.
Zhao T, Dai A (2016) Uncertainties in historical changes and future projections of drought. Part II: model-simulated historical and future drought changes. Climatic Change. DOI: 10.1007/s10584-016-1742-x.

>> As suggested by the reviewer, we have added the reviews on the projected drought changes as follows.

Page 2: *"Zhao and Dai (2015; 2016) assessed the self-calibrated PDSI (sc-PDSI) with multiple CMIP3 and CMIP5 model projections at the globe scale and showed that the drought frequency and area increased with increasing sc-PDSI, even under low to moderate emission scenarios."*

4. The title should be clarified the drought index of the SPEI because this manuscript only focuses on the SPEI but not on other drought index.

>> As suggested by the reviewer, we have revised the title from *"Effects of different reference periods on drought index estimations for 1901-2014"* to *"Effects of different reference periods on drought index (SPEI) estimations from 1901-2014"*.

5. Line 21, P1, 'limited to' should be 'limited by'.

>> As suggested, we have made this correction.

6. Line 20-21, P1, "Although" has been used in this sentence, why is "nevertheless" still used?

>> As suggested, we have deleted this text.

7. Line 31, P1, 'stated', is not 'state'.

>> As suggested, we have corrected this term.

8. Line 31, P2, 'This study' would be revised as 'our study' to clarify the following description is the main findings obtained from this manuscript.

>> As suggested, we have revised this text.

9. 'Data and method' is better for the title of the section 2.

>> As suggested, we have revised the title of section 2.

10. There are large gaps for the English writing for this manuscript.

>> Professional English editors have reviewed the manuscript thoroughly, and the editing certificate is attached below.

[Figure]

**EDITORIAL CERTIFICATE**

This document certifies that the manuscript listed below was edited for proper English language, grammar,
punctuation, spelling, and overall style by one or more of the highly qualified native
English speaking editors at American Journal Experts.

**Manuscript title:**

Effects of different reference periods on drought index (SPEI) estimations from 1901-2014

**Authors:**

Myoung-Jin Um, Yeonjoo Kim, Daeryong Park, Jeongbin Kim

**Date Issued:**

August 10, 2017

**Certificate Verification Key:**

C4A4-9799-5038-D726-A0D5

[Figure]

This certificate may be verified at www.aje.com/certificate. This document certifies that the manuscript listed above was edited for proper English language, grammar, punctuation, spelling, and overall style by one or more of the highly qualified native English speaking editors at American Journal Experts. Neither the research content nor the authors' intentions were altered in any way during the editing process. Documents receiving this certification should be English-ready for publication; however, the author has the ability to accept or reject our suggestions and changes. To verify the final AJE edited version, please visit our verification page. If you have any questions or concerns about this edited document, please contact American Journal Experts at support@aje.com.

American Journal Experts provides a range of editing, translation and manuscript services for researchers and publishers around the world. Our top-quality PhD editors are all native English speakers from America's top universities. Our editors come from nearly every research field and possess the highest qualifications to edit research manuscripts written by non-native English speakers. For more information about our company, services and partner discounts, please visit www.aje.com.

**Anonymous Reviewer # 4 (Report #2)**

I am providing comments on the manuscript entitled "Effects of different reference periods on drought index estimations for 1901-2014" submitted to HESS. In my opinion, the manuscript touches an interesting and important subject which is the effect of the reference period on drought analysis. Generally, the manuscript is well organized, concise and to the point. However, I noticed some editorial issues such as verb tense that need to be take care of before final submission.

>> Professional English editors have reviewed the manuscript thoroughly, and the editing certificate is attached below.

[Figure]

The manuscript argues that drought analysis is sensitive to the choice of reference period and support this claim through some interesting analysis. This is the strong aspect of the manuscript; however, it does not inform those who want to perform drought analysis on what they should do. I feel the manuscript establishes the challenge or question well but it is relatively weak when it comes to providing some solutions or answers. Adding such additional discussion will increase the influence of the manuscript.

>> As suggested, we have added discussion of our findings and their implications, as well as potential topics of future studies.

[revised manuscript text omitted]